# Subcortical circuits mediate communication between primary sensory cortical areas in mice

Michael Lohse [1,2 ✉], Johannes C. Dahmen [1], Victoria M. Bajo [1] & Andrew J. King [1 ✉]

Integration of information across the senses is critical for perception and is a common property of neurons in the cerebral cortex, where it is thought to arise primarily from corticocortical connections. Much less is known about the role of subcortical circuits in shaping the multisensory properties of cortical neurons. We show that stimulation of the whiskers causes widespread suppression of sound-evoked activity in mouse primary auditory cortex (A1). This suppression depends on the primary somatosensory cortex (S1), and is implemented through a descending circuit that links S1, via the auditory midbrain, with thalamic neurons that project to A1. Furthermore, a direct pathway from S1 has a facilitatory effect on auditory responses in higher-order thalamic nuclei that project to other brain areas. Cross-modal corticofugal projections to the auditory midbrain and thalamus therefore play a pivotal role in integrating multisensory signals and in enabling communication between different sensory cortical areas.

[1] Department of Physiology, Anatomy, and Genetics, University of Oxford, Oxford, UK. [2] Present address: Sainsbury Wellcome Centre, London, UK.
✉email: michael.lohse@dpag.ox.ac.uk; andrew.king@dpag.ox.ac.uk

 1

Having multiple sensory systems, each specialized for the transduction of a different type of physical stimulus, maximizes our ability to gather information about the external world. Furthermore, when the same event or object is registered by more than one sense, as is often the case, our chances of detecting and accurately evaluating its biological significance dramatically increase[1]. Unlike audition and vision, the sense of touch informs an organism exclusively about objects in its immediate vicinity. This is particularly important in animals that rely on their whiskers for detecting the presence and location of objects as they explore their surroundings[2]. Inputs from the whiskers can enhance sound-induced defensive behavior[3] and neural mechanisms that give precedence to the processing of somatosensory information over cues from other modalities are likely to be advantageous to the organism's survival.

Apart from specialized subcortical premotor nuclei, such as the superior colliculus, it is widely assumed that multisensory processing is most prevalent at the level of the cerebral cortex[1,4]. Evidence for multisensory convergence has been found in nearly all cortical areas, including the primary sensory cortices. In the primary auditory cortex (A1), for example, visual or tactile stimuli can modulate acoustically-driven activity, most commonly by suppressing responses to sound in both awake and anesthetized animals[5–8]. Suppression of sound-evoked activity in auditory cortical neurons by somatosensory inputs likely provides a mechanism for prioritizing the processing of tactile cues from nearby objects that require urgent attention.

The circuitry underlying crossmodal influences on processing in early sensory cortical areas is poorly understood. Because visual, auditory and somatosensory cortices innervate each other and connect with higher-level, association areas[5,7,9–14], most studies have focused on the role of intracortical circuits in multisensory integration[15–18]. This, however, ignores the potential contribution of ascending inputs from the thalamus, which may also provide a source of multisensory input to primary cortical areas, such as A1[11,19–22], or the possibility that early sensory cortical areas may communicate via a combination of corticofugal and thalamocortical pathways[23,24].

In this paper, we investigate whether subcortical sensory circuits play a role in shaping multisensory processing in cortex. We show that somatosensory inputs exert a powerful influence on processing in the auditory system, which is independent of brain state and takes the form of divisive suppression in the auditory thalamus and cortex. Dissecting the underlying circuitry, we found that this suppression originates in the primary somatosensory cortex (S1) and can be implemented via S1-recipient neurons in the auditory midbrain, which inhibit sound-driven activity in the auditory thalamocortical system. We also show that a parallel crossmodal corticothalamic pathway from S1 to the medial sector of the auditory thalamus allows for somatosensory facilitation of auditory responses in thalamic neurons that do not project to the auditory cortex. These results demonstrate that the auditory midbrain and thalamus have essential roles in integrating somatosensory and auditory inputs and in mediating communication between cortical areas that belong to different sensory modalities.

## Results

### Somatosensory influences on primary auditory cortex.
Because variable effects of tactile stimulation have been reported on the activity of neurons in the auditory cortex of different species[6,7,25–27], we recorded extracellular activity in A1 of awake mice, while presenting tones and simultaneously deflecting the whiskers (Fig. 1a). We consistently found that concurrent whisker stimulation reduced auditory responses (Fig. 1a-c), demonstrating

widespread suppression of auditory activity in A1. Furthermore, assessment of the input-output responses across all tones presented, normalized to the firing rate at each neuron's best frequency (BF), revealed that this suppression was stimulus-specific and of a divisive nature, with strong effects around the BF and negligible effects for off-BF responses that were closer to baseline activity (Fig. 1d, e).

To test whether this somatosensory suppression is mediated by local inhibitory interneurons, potentially targeted by direct cortico-cortical connections from S1 to A1, we performed 2-photon calcium imaging of inhibitory interneurons (VGAT-positive cells) in A1 of awake mice (Supplementary Fig. 1a). We found that the auditory responses of inhibitory neurons in A1 were also suppressed by whisker stimulation ($P < 0.001$, $n = 514$, 3 mice; Supplementary Fig. 1b, c). This suggests that whisker-stimulation-induced suppression in A1 is unlikely to reflect increased activity of local interneurons, as has been demonstrated for the suppressive effects of motor-related signals on auditory cortical activity[28].

Furthermore, to assess whether the suppression could be attributed to non-sensory influences, such as stimulus-triggered movements of the whiskers, changes in attention, or arousal, we also made electrophysiological recordings from A1 of anesthetized mice and again observed a stimulus-dependent suppression of auditory responses, with the strongest effects around the units' BF (Fig. 2a–d). These findings indicate that the suppression of auditory responses by whisker stimulation is caused by an interaction between the somatosensory and auditory system that operates robustly across different brain states.

### Somatosensory influences on auditory thalamus.
To investigate the circuitry underlying this extensive modulation of auditory cortical processing, we first set out to determine whether the activity of subcortical auditory neurons is similarly affected by whisker stimulation. To maintain control over brain state and avoid self-generated movement of the whiskers during sensory stimulation, we carried out the majority of the circuit dissection experiments in anesthetized mice (unless specified otherwise). We found no evidence for somatosensory-auditory interactions in the central nucleus of the inferior colliculus (CNIC) (change in BF response, $P > 0.05$, $n = 58$ (2 mice); Supplementary Fig. 2) and therefore focused on the medial geniculate body (MGB), the main thalamic gateway to the auditory cortex. We recorded from neurons in the lateral region of the MGB, including both the lemniscal ventral division (MGBv) and the non-lemniscal dorsal division (MGBd) (Fig. 2 and Supplementary Fig. 3). Whisker stimulation suppressed responses to noise and to tones near the BF of neurons in both MGBv and MGBd (Fig. 2a, e–g and Supplementary Fig. 4). As in the cortex, this suppression took the form of a divisive scaling of the sound-evoked response (Fig. 2d, g). Given that very similar divisive suppression was induced by whisker stimulation in lemniscal MGBv and non-lemniscal MGBd, we chose to analyze the data from these two regions together when investigating somatosensory modulation of auditory thalamus. Somatosensory influences on auditory responses were also found in MGBv and MGBd of awake, head-fixed mice, with the largest suppressive effects again being found close to BF (change in BF response, $P_{60dB\_SPL} = 6.6 \times 10^{-7}$, $P_{80dB\_SPL} = 0.01$, $n = 157$, 5 mice, Supplementary Fig. 5).

The medial section of the auditory thalamus contains several subdivisions, medial MGB (MGBm), the posterior intralaminar nucleus (PIN), and the suprageniculate nucleus (SGN), which are anatomically distinct from the MGBv and MGBd[29–31]. The effects of whisker stimulation were very similar across each of these medial thalamic regions and were therefore analyzed together.

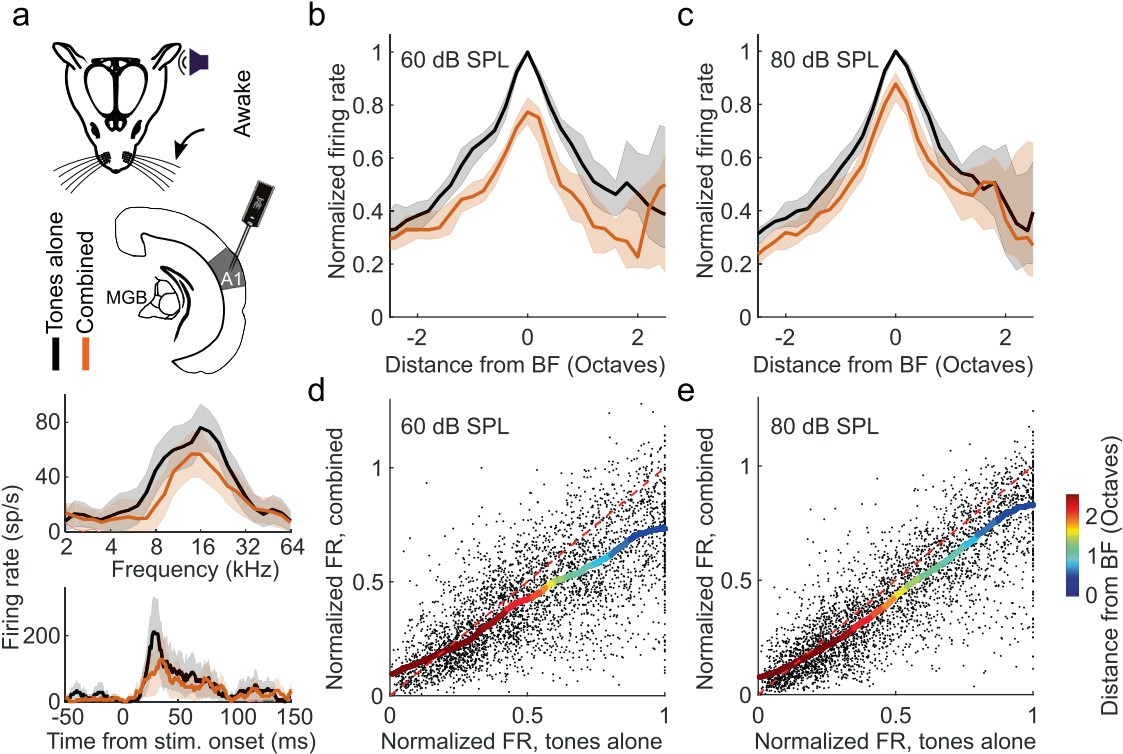

**Fig. 1 Somatosensory suppression of neurons in primary auditory cortex (A1) of awake, head-fixed mice. a** Top: Schematic of recording setup. MGB, medial geniculate body. Bottom: Example frequency response profiles and PSTHs of best frequency (BF) responses from a unit recorded in A1 of an awake, passively listening mouse, illustrating tone responses (80 dB SPL) with (orange) or without (black) concurrent whisker stimulation. sp/s, spikes per second. Median frequency response profiles for tones presented at 60 (**b**) and 80 dB SPL (**c**) across units recorded in A1 of awake mice (60 dB SPL change in BF response: $P = 1.7 \times 10^{-12}$, $n = 140$, two-sided Wilcoxon signed-rank test; 80 dB SPL change in BF response: $P = 1.2 \times 10^{-13}$, $n = 140$, two-sided Wilcoxon signed-rank test). Relationship between normalized firing rate (FR) for all A1 units (black dots) for tones presented at 60 (**d**) or 80 dB SPL (**e**) across all frequencies either with ("combined") or without ("tones alone") whisker stimulation. Thick multi-colored lines show the running median of this relationship (window: 0.1 normalized firing rate), and the colors denote the distance from BF. The diagonal dashed red line is the line of equality. A larger distance between the multi-colored line and the diagonal line at the blue end than at the red end indicates divisive scaling. The shaded area indicates the 95% confidence intervals of the means (**a** frequency response profiles), the s.e.m. (**a** PSTHs) or the 95% nonparametric confidence intervals of the median (**b**, **c**). $n = 140$ (4 mice).

We found that over a quarter (27.6%, a higher proportion than the 5% expected by chance, $P < 0.001$, binomial test) of noise-responsive units in the MGBm/PIN and SGN were directly driven ($P < 0.05$, one-sided $t$ test) by whisker inputs alone (Fig. 3a, c, i). The responses of individual units to noise (Fig. 3a, b, d, i) or tones (Fig. 3e, f, g, h, i) could be either facilitated or suppressed when combined with whisker input. Units in which responses to tones were facilitated exhibited an increase in firing rate across all sound frequencies tested, indicative of additive scaling (Fig. 3e, f), whereas suppressed units, similar to those in MGBv/d and cortex, showed divisive scaling (Fig. 3g, h). Thus, neurons in the medial section of the auditory thalamus were influenced by whisker stimulation in a much more heterogeneous fashion than neurons in the lateral MGB (Fig. 3j). We found similarly diverse modulations of auditory responses in MGBm/PIN and SGN in awake, head-fixed mice (7/52 units had significantly ($P < 0.05$) facilitated BF responses, and 5/52 units had significantly ($P < 0.05$) suppressed BF responses; Supplementary Fig. 6).

Because our results suggest a functional segregation for somatosensory-auditory interactions in the MGB between the lateral nuclei (MGBv and MGBd) and the medial nuclei (MGBm, PIN, SGN) (Fig. 3j), we considered MGBv and MGBd as one functional module (MGBv/d), and MGBm, PIN and SGN as another functional module (MGBm/PIN/SGN) for the analysis of the circuitry underlying the effects of whisker stimulation on neural responses in the auditory thalamus.

**Auditory thalamocortical neurons are suppressed by whisker stimulation.** Whisker-stimulation induced suppression of auditory activity is therefore present subcortically, particularly in the MGBv and MGBd, two auditory thalamic subdivisions with massive thalamocortical projections. This suggests that cortical neurons may receive signals in which acoustic and somatosensory information have already been integrated. To investigate whether MGB neurons do indeed relay a whisker-modulated signal to auditory cortex, we expressed the calcium indicator GCaMP6m in the entire auditory thalamus and measured calcium transients in thalamocortical boutons in layer 1 of the auditory cortex (Fig. 4a, b). Layer 1 of the mouse auditory cortex tends to receive more diverse thalamic inputs than layers 3b/4. In A1, for example, layer 1 combines dense projections from MGBv[31,32], including collaterals of axons innervating layers 3b/4[31], with projections from other structures, such as MGBm[31] and the lateral posterior nucleus of the thalamus[21]. By imaging thalamocortical axons that terminate in layer 1, we should therefore sample the effects of somatosensory influences on sound-evoked activity transmitted from both lateral and medial regions of the auditory thalamus. We found that whisker stimulation had a suppressive effect on the majority of thalamocortical bouton responses to both noise (Fig. 4c, d) and tones (Fig. 4e, f). Similar to neurons in MGBv, MGBd and auditory cortex, frequency-tuned thalamocortical boutons exhibited divisive scaling with the largest response reduction at BF (Fig. 4e, f). We did not find any auditory

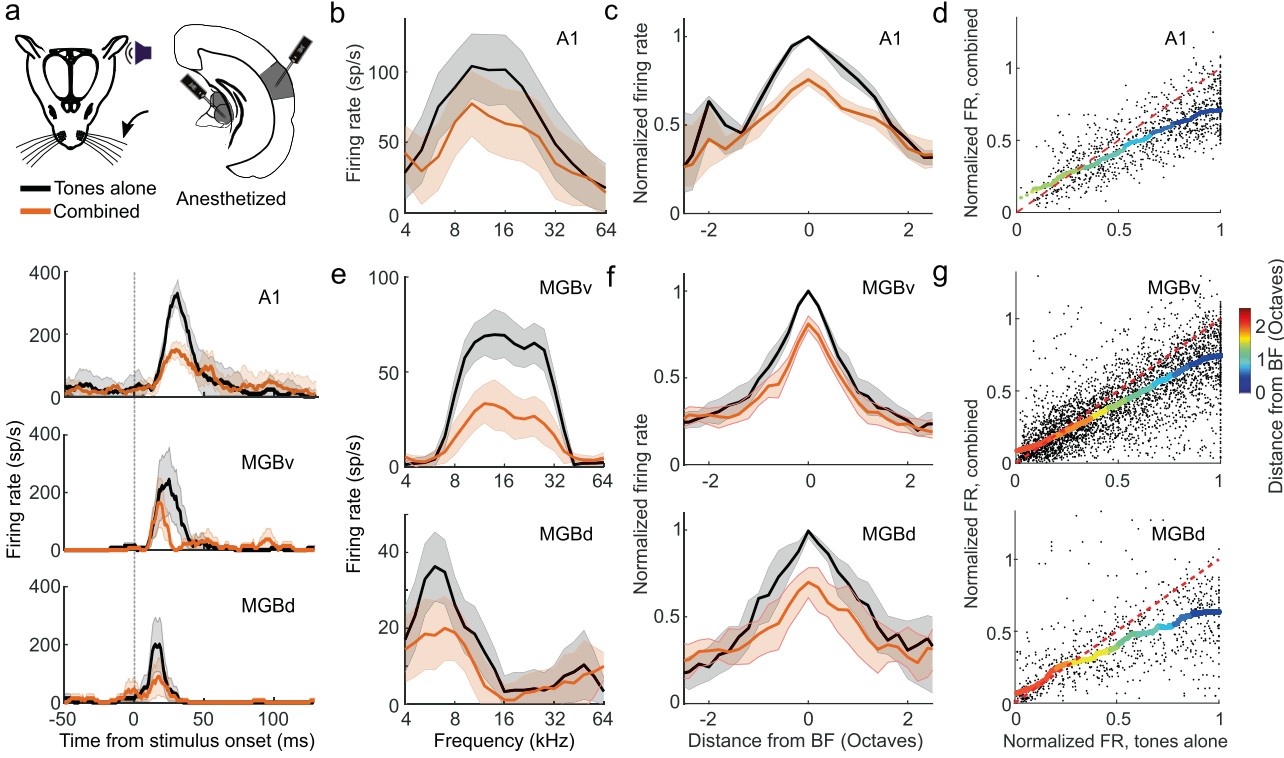

**Fig. 2 Divisive scaling of frequency tuning by somatosensory inputs in the primary auditory cortex (A1) and the ventral and dorsal divisions of the medial geniculate body (MGBv and MGBd) of anesthetized mice. a** Top: Schematic of recording setup. Bottom: Example PSTHs illustrating best frequency (BF) responses with (orange) or without (black) concurrent whisker stimulation from units in A1, MGBv, and MGBd. **b** Example frequency response profiles with or without concurrent whisker stimulation from the same A1 unit. sp/s, spikes per second. **c** Median tuning curve across units recorded in A1 (change in BF response: $P = 3.6 \times 10^{-12}$, $n = 77$, two-sided Wilcoxon signed-rank test). **d** Relationship between normalized firing rate (FR) for all A1 units (black dots) for tones across all frequencies presented either with ("combined") or without ("tones alone") whisker stimulation. **e** Frequency response profiles from the same MGB (top: MGBv, bottom: MGBd) units depicted in **a** either with or without concurrent whisker stimulation. **f** Median frequency response profiles across units recorded in MGBv (top, change in BF response: $P = 2.5 \times 10^{-16}$, $n = 145$, two-sided Wilcoxon signed-rank test) and MGBd (bottom, change in BF response, $P = 1.3 \times 10^{-4}$, $n = 31$, two-sided Wilcoxon signed-rank test) with or without concurrent whisker stimulation. **g** Relationship between normalized firing rate (FR) for all units (black dots) recorded in the MGBv (top) and MGBd (bottom) for tones across all frequencies presented either with ("combined") or without ("tones alone") whisker stimulation. Thick multi-colored lines show the running median of this relationship (window: 0.1 normalized firing rate), and the colors denote the distance from BF. The diagonal dashed red line is the line of equality. A larger distance between the multi-colored line and the diagonal line at the blue end than at the red end indicates divisive scaling. The shaded area indicates the s.e.m. (**a**), the 95% confidence intervals of the means (**b**, **e**), or the 95% nonparametric confidence intervals of the median (**c**, **f**). $n_{A1} = 77$ (4 mice), $n_{MGBv} = 145$ (9 mice); $n_{MGBd} = 31$ (9 mice). See Supplementary Fig. 4 for similar results in awake, head-fixed mice.

thalamocortical boutons that were driven by whisker stimulation alone or whose sound responses were facilitated by whisker stimulation. This supports the hypothesis that only somatosensory suppression of auditory activity is projected to the auditory cortex, whereas the facilitation observed in the medial sector of the auditory thalamus is not. Although we cannot rule out the possibility that MGBm axons carrying somatosensory drive and facilitation may terminate in the deep layers of A1, which were not imaged here, our electrophysiological data do not reveal evidence for multisensory facilitation in those layers (Figs. 1, 2; Supplementary Fig. 5).

**Primary somatosensory cortex mediates suppression of the auditory thalamus.** To determine whether S1 is involved in whisker-stimulation induced suppression of the auditory thalamocortical system, we recorded neuronal activity in the MGB of VGAT-ChR2-YFP mice whilst silencing S1 optogenetically (Fig. 5a and Supplementary Fig. 7). Silencing S1 did not affect spontaneous activity or tone-evoked auditory thalamic responses ($P > 0.05$, $n_{MGBv/d} = 59$, 3 mice; $n_{MGBm/PIN/SGN} = 84$, 3 mice; Supplementary Fig. 7), but significantly reduced the capacity of

whisker stimulation to suppress the BF responses of neurons in both MGBv and MGBd (Fig. 5b,c). Thus, S1 is a critical part of the circuitry mediating the somatosensory control of auditory thalamocortical responses.

Silencing S1 did not affect the responses of neurons in the medial sector of the auditory thalamus ($P_{suppression} > 0.05$, $n = 11/84$ units; $P_{facilitation} > 0.05$, $n = 10/84$ units; Supplementary Fig. 8). S1 is thus necessary for somatosensory suppression in the MGBv/d, but not for somatosensory modulation in the MGBm/PIN/SGN. That S1 activation is also sufficient for the suppression of auditory thalamocortical responses was revealed when we optogenetically activated infragranular cells in S1 via the red-shifted opsin ChrimsonR and measured calcium transients in thalamocortical boutons (Fig. 5d, e). Optogenetic S1 activation suppressed their responses to noise bursts (Fig. 5f, g) and thus replicated the previously observed whisker-induced suppression of auditory thalamocortical boutons.

**Auditory cortex does not mediate somatosensory influences on auditory thalamus.** Our 2-photon imaging data, described above, suggest that S1 does not suppress A1 activity by targeting local

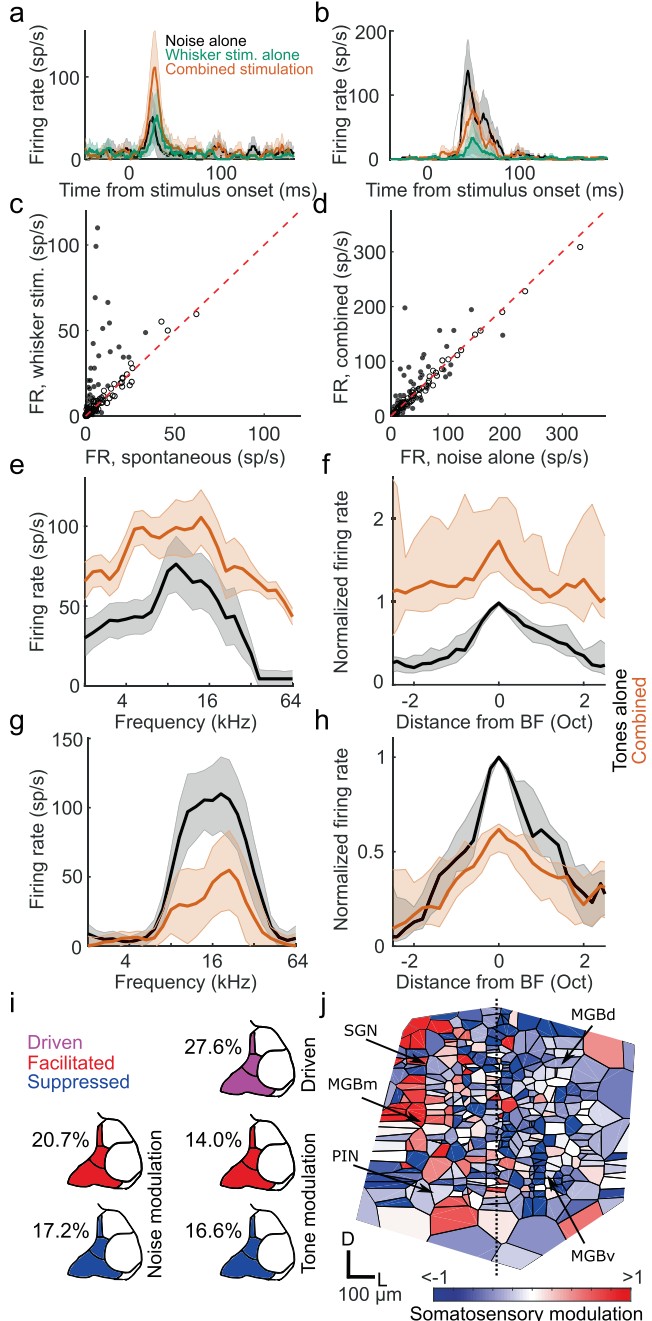

**Fig. 3 Diverse somatosensory influences on neurons in the medial auditory thalamus.** Example PSTHs of responses to broadband noise recorded in MGBm/PIN/SGN with (orange) and without (black) concurrent whisker stimulation, as well as to whisker stimulation alone (green), showing somatosensory facilitation (**a**) and suppression (**b**) of the auditory response, respectively (defined using a criterion of $P < 0.05$, two-side $t$ test). **c** Summary of responses (firing rate, FR) to whisker stimulation alone vs spontaneous activity in the medial sector of the auditory thalamus. Filled circles indicate units driven by somatosensory stimulation ($P < 0.05$, two-sided $t$ test). **d** Summary of responses to broadband noise combined with or without concurrent whisker stimulation. Filled circles indicate significantly ($P < 0.05$, two-sided $t$ test) modulated units ($n = 116$, 8 mice). **e** Example frequency response profiles for tones with (orange) and without (black) concurrent whisker stimulation for a unit showing crossmodal facilitation ($P < 0.05$, two-sided $t$ test). **f** Summary frequency response profiles of units with significantly facilitated responses at their best frequency (BF). **g**, **h** Same as **e**, **f** for units with significantly suppressed BF responses. $n_{facilitated} = 32$, $n_{suppressed} = 27$, 12 mice. Oct, octaves; sp/s, spikes per second. The shaded area indicates 95% confidence intervals of the mean (**a**, **b**, **e**, **g**) or nonparametric confidence intervals of the medians (**f**, **h**), respectively. **i** Percentage of neurons in the MGBm/PIN and SGN significantly ($P < 0.05$, one-sided $t$ test) driven by somatosensory input, or showing significant modulation ($P < 0.05$, two-sided $t$ test) of the responses to noise or tones at BF when combined with somatosensory input. **j** Voronoi diagram illustrating the location across the auditory thalamus (collapsed in the rostro-caudal plane) of all tuned neurons that were modulated by somatosensory stimulation. Each patch represents the location of one extracellularly recorded thalamic unit ($n = 369$, 14 mice) and is color-coded for the type and strength of somatosensory modulation (red, facilitation; blue, suppression). D, dorsal; L, lateral; MGBd, Medial Geniculate Body dorsal division; MGBm, MGB medial division; MGBv, MGB ventral division; PIN, posterior intralaminar nucleus; SGN, supragenicular nucleus. See supplementary Fig. 6 for similar results in awake, head-fixed mice.

## S1 projection neurons account for auditory thalamic facilitation.

To investigate whether a direct corticothalamic projection[24,33,34] exists that could mediate somatosensory control over auditory thalamus, we performed viral tracing experiments in S1 corticothalamic neurons. These revealed that a projection does indeed exist, which originates from RBP4-expressing layer 5 neurons in S1 and densely innervates the medial sector of auditory thalamus (Fig. 6a–c), particularly the PIN (Fig. 6b, c). Optical stimulation of these S1 layer 5 neurons significantly altered the spontaneous firing rate of more than a third of recorded units (Fig. 6d, e), suggesting a direct excitatory pathway from S1 to the medial auditory thalamus. Activation of this pathway also replicated the additive scaling of the frequency response profiles of auditory neurons recorded in this region of the auditory thalamus (Fig. 6f, g) that we observed when combining sounds and whisker stimulation.

Although these findings are consistent with a facilitatory influence of layer 5 projection neurons in S1 on neurons in the medial auditory thalamus, selective stimulation of the RBP4-expressing neurons did not induce suppression of the sound-evoked responses of neurons recorded in the MGBv and MGBd (Supplementary Fig. 10). This result can be readily accounted for given the generally excitatory nature of corticofugal projections and the predominantly medial termination pattern of this particular pathway, as well as the relative paucity of GABAergic interneurons in the rodent MGB[35]. Nevertheless, the lack of effect of stimulating S1 RBP4-expressing neurons on the sound-evoked responses of neurons recorded in the lateral auditory thalamus contrasts with the reduced influence of whisker stimulation on

inhibitory interneurons (Supplementary Fig. 1). However, to rule out the possibility that descending auditory corticothalamic inputs contribute to the effects of whisker stimulation on the MGB, we recorded from the auditory thalamus while optogenetically silencing A1. Silencing auditory cortex strongly decreased both spontaneous activity ($P_{MGBv/d} < 0.001$, $n_{MGBv/d} = 59$, 3 mice; $P_{MGBm/PIN/SGN} < 0.001$, $n_{MGBm/PIN/SGN} = 84$, 3 mice; Supplementary Fig. 7) and sound-evoked responses in auditory thalamic neurons ($P_{MGBv/d} < 0.001$, $n_{MGBv/d} = 59$, 3 mice; $P_{MGBm/PIN/SGN} < 0.001$, $n_{MGBm/PIN/SGN} = 84$, 3 mice; Supplementary Fig. 7). However, silencing A1 did not alter the modulatory effects of whisker stimulation on the responses of neurons in either the MGBv/d ($P > 0.05$, $n_{MGBv/d} = 59$, 3 mice) or the medial sector of the auditory thalamus ($P > 0.05$, $n_{MGBm/PIN/SGN} = 84$, 3 mice; Supplementary Fig. 9). This finding therefore indicates that an indirect corticocorticothalamic pathway is not responsible for the effects of S1 on neuronal activity in the auditory thalamus.

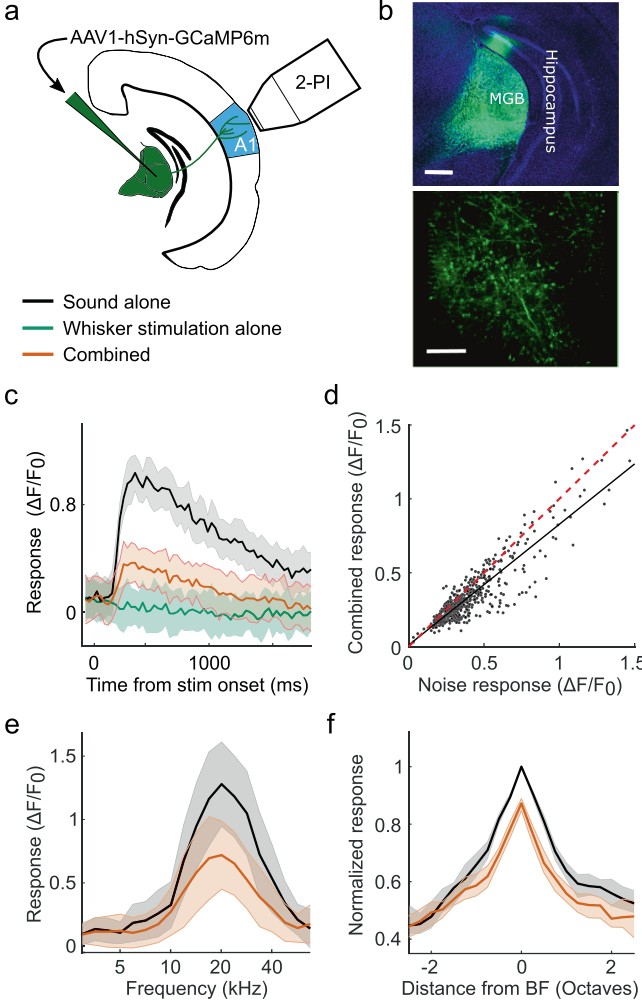

— Sound alone
— Whisker stimulation alone
— Combined

**Fig. 4 Thalamic inputs to auditory cortex are suppressed by whisker stimulation. a** Schematic of recording setup. 2-PI, 2-photon imaging; A1, primary auditory cortex. **b** Top: Confocal image of GCaMP6m expression in the auditory thalamus. Scale bar, 400 μm. Bottom: In vivo 2-photon image of thalamocortical boutons in layer 1 of the auditory cortex. Scale bar, 20 μm. **c** Calcium response of an example thalamic bouton in layer 1 responding to broadband noise with (orange) or without (black) concurrent whisker deflection, as well as to whisker deflection alone (green). **d** Summary of responses to noise alone vs combined noise plus whisker deflection in all noise-responsive thalamocortical boutons ($P = 2.4 \times 10^{-44}$, $n = 512$, 3 mice, two-sided Wilcoxon signed-rank test). The red dashed line indicates the line of equality. The black solid line indicates the least squares linear fit. **e** Frequency response profiles with (orange) and without (black) whisker deflection from an example thalamocortical bouton. **f** Median frequency response profiles across all frequency tuned boutons (change in response at best frequency (BF): $P = 7.0 \times 10^{-13}$, $n = 310$, 3 mice, two-sided Wilcoxon signed-rank test). Shaded area indicates the 95% confidence intervals of the means (**c**,**e**), or the 95% nonparametric confidence intervals of the median (**f**).

those responses when S1 was silenced optogenetically. This therefore implies the existence of another pathway by which S1 neurons can influence auditory processing in this part of the thalamus.

**A corticocollicular pathway for somatosensory thalamic suppression.** The final objective was to identify the source of inhibition mediating S1-dependent suppression of neuronal activity in the auditory thalamus. One major source of inhibitory input to

the MGB, and a structure that has previously been implicated in crossmodal thalamic processing, is the thalamic reticular nucleus (TRN)[36]. By optogenetically silencing the auditory sector of TRN (AudTRN) during tone presentation, we found that this part of the thalamus modulates the excitability of MGB neurons (Fig. 7a–c). Surprisingly, however, we did not find any evidence that AudTRN neurons play a role in mediating somatosensory suppression of the MGB in anesthetized mice (Fig. 7d, e). Although we cannot rule out the possibility that TRN neurons may additionally contribute to crossmodal modulation in awake, behaving animals, our results suggest that they are not responsible for somatosensory suppression of neurons in MGBv/d, which occurs in both awake and anesthetized mice.

Inhibitory input to the MGB can also arrive from extra-thalamic sources, including the IC[37–39], which provides its major source of ascending input. Although whisker stimulation had no effect on auditory responses in the CNIC (Supplementary Fig. 2), descending inputs from the somatosensory cortex have been reported to target modular zones containing GABAergic neurons within the lateral shell of the mouse IC[37], suggesting a possible route by which whisker inputs could influence auditory processing. To examine this possibility, we recorded from neurons ($n = 94$, 2 mice) in the lateral cortex of the IC (LCIC) and found that a subset of frequency-tuned neurons was driven by whisker stimulation alone (17%, $P < 0.05$, one-sided $t$ test) and/or facilitated by whisker stimulation (7.5%, $P < 0.05$, two-sided $t$ test). Another subset of LCIC neurons had their auditory responses suppressed by whisker stimulation (9.5%, $P < 0.05$, $t$ test; Supplementary Fig. 11). We also employed an anterograde trans-synaptic viral tagging approach[40] in which AAV1-hSyn-cre was injected into auditory cortex of GCaMP6f reporter mice in order to largely restrict GCaMP labeling to the IC shell, the primary target of descending inputs from auditory cortex[41]. Using two-photon calcium imaging, we found that the BF responses of neurons in the optically accessible dorsal cortex of IC were suppressed by concurrent whisker stimulation ($P < 0.001$, $n = 232$ cells, 2 mice; Supplementary Fig. 12). Thus, the responses of IC shell neurons are modulated by somatosensory inputs, with the suppressive effects presumably reflecting either reduced signals from auditory corticocollicular neurons during whisker stimulation or the action of inhibitory circuits within the IC.

To investigate more directly the IC circuitry mediating these crossmodal interactions, we injected AAV1-hSyn-cre into S1 and AAV1-CAG-FLEX-tdtomato into the lateral IC of transgenic mice that expressed YFP in GABAergic neurons. This allowed us to show that S1 directly targets GABAergic LCIC neurons and that these neurons project to the auditory thalamus (Supplementary Fig. 13). Furthermore, in order to manipulate the activity of S1-recipient neurons, we induced expression of channelrhodopsin-2 in these S1-recipient IC neurons (Fig. 7f, g). Activating them resulted in suppression of auditory responses both in MGBv/d (Fig. 7h–j) and the medial auditory thalamus (Fig. 7k–m). This demonstrates that S1 can exert suppressive control over auditory thalamic processing via a corticocolliculothalamic pathway, in addition to its facilitatory influence via a direct crossmodal corticothalamic pathway (Fig. 8).

## Discussion

We found that somatosensory inputs have diverse and anatomically specific effects on auditory thalamocortical processing in mice. We identified two separate corticofugal pathways (Fig. 8), which both originate in S1 but exert opposing influences over the auditory thalamus. First, a crossmodal descending pathway via the auditory midbrain can mediate somatosensory divisive suppression in the auditory thalamocortical system. Second, a direct

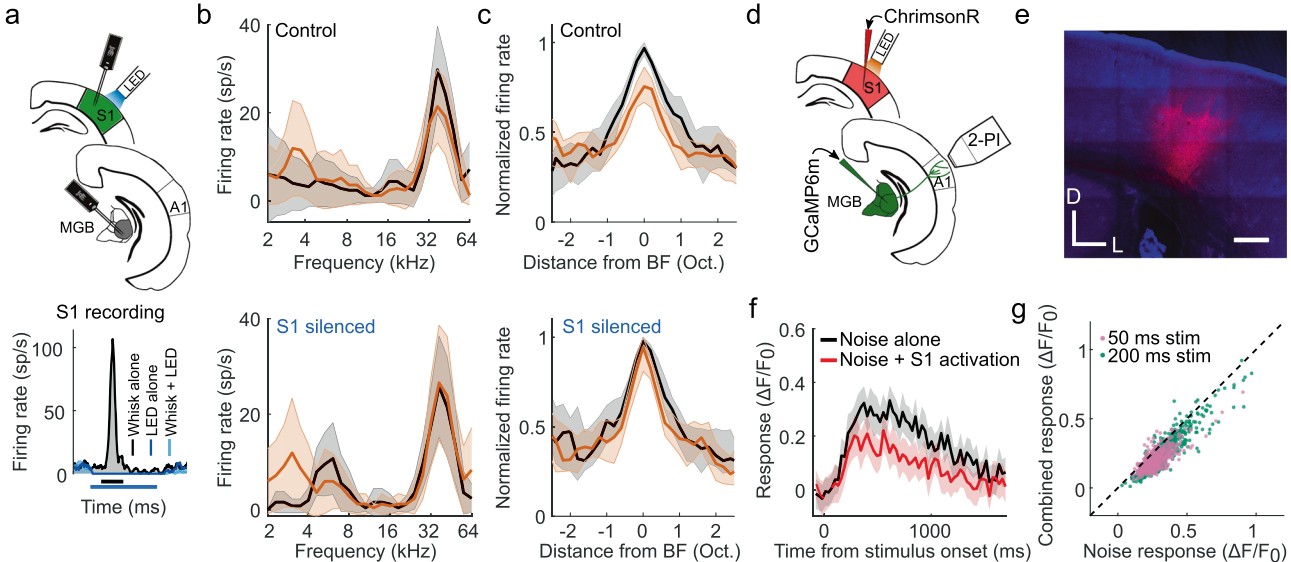

**Fig. 5 S1 mediates somatosensory suppression of auditory thalamocortical axons. a** Top: Schematic of optogenetic targeting of primary somatosensory cortex (S1) in VGAT-ChR2 mice and electrophysiological recording setup. Bottom: Example PSTHs of a unit recorded in S1, demonstrating the effect of optogenetic silencing of somatosensory cortex on spontaneous activity and whisker-stimulation evoked responses. Bars below the x-axis indicate timing of whisker stimulation (black) and photostimulation for silencing S1 (blue). **b** Frequency response profiles of an example MGBv unit based on tone responses with (orange) and without (black) concurrent whisker stimulation during the control condition (top) and when S1 was silenced (bottom). **c** Median frequency response profiles of all units recorded in MGBv/d with (orange) and without whisker deflection (black) during the control condition (top) and when S1 was silenced (bottom). Because of the comparable effects of whisker stimulation on the responses of neurons in the MGBv and MGBd, we analyzed these interactions by combining data from these two regions of the auditory thalamus. The suppressive effect of whisker stimulation on the response at best frequency (BF) of MGBv/d neurons was reduced following S1 silencing ($P = 0.01$, $n = 59$, 3 mice, two-sided Wilcoxon signed-rank test). **d** Schematic of experimental setup for combined 2-photon thalamocortical bouton imaging (2-PI) with optogenetic activation of S1. **e** Confocal image showing expression of ChrimsonR-tdTomato in infragranular layers of S1. Scale bar, 300 μm. D, dorsal; L, lateral. **f** Calcium response of an example thalamic bouton in layer 1 of the auditory cortex, illustrating suppression of the response to a 50 ms noise burst by optogenetic S1 stimulation. Shading indicates 95% confidence intervals around the mean. The 3rd and 4th imaging frames of the S1 stimulation condition displayed a large light artifact from the LED and have therefore been removed. **g** Summary plot of responses to noise alone or noise combined with infragranular S1 stimulation for all noise-responsive boutons. Purple and green points indicate responses to 50 ms and 200 ms noise stimulation, respectively. $n_{50ms} = 539$, 8 imaging fields, 1 mouse; $n_{200ms} = 652$, 7 imaging fields, 2 mice. Shaded area indicates the 95% confidence intervals of the means (**b**, **f**), or the 95% nonparametric confidence intervals of the median (**c**). A1, primary auditory cortex; MGB, medial geniculate body; MGBd, MGB dorsal division; MGBv, MGB ventral division; sp/s, spikes per second.

corticothalamic pathway targets the medial sector of auditory thalamus, through which S1 drives spiking activity and facilitates neuronal responses that do not appear to be transmitted to the auditory cortex. These findings, therefore, reveal an unexpected role for corticofugal projections to both the auditory midbrain and thalamus in shaping the multisensory properties of auditory cortical and other downstream neurons and in enabling communication between different cortical areas.

**Auditory cortex inherits multisensory signals from the thalamus.** Although spiking responses to visual or somatosensory stimuli have been found in different parts of auditory cortex, the commonest type of crossmodal interaction reported is a modulation of sound-evoked responses by otherwise ineffective sensory stimuli[5–8,18,25,26,42,43]. In line with our results, crossmodal suppressive interactions are frequently observed, both in rodents[6,21,27] and other species[5,7,8,44]. Because direct connections exist between sensory cortical areas[5,7,9–13], the search for the origin of these multisensory cortical responses has focused principally on other cortical areas. For example, somatosensory cortical responses in cats can be suppressed by sound or by electrical activation of the auditory anterior ectosylvian sulcal field and this crossmodal modulation is blocked by local application of a GABA receptor antagonist[45]. Furthermore, in mice, optogenetic stimulation of A1 corticocortical projections can modulate the activity[15,16] and stimulus selectivity[16] of

neurons in primary visual cortex via local inhibitory circuits. Our data suggest, however, that a local A1 circuit is not responsible for the effects of whisker stimulation on auditory responses since both excitatory and inhibitory neurons were suppressed.

While corticocortical connections can contribute to multisensory interactions, we show that non-auditory influences on auditory cortical processing are also inherited from the thalamus. Anatomical studies have emphasized the potential contribution to multisensory responses in the auditory cortex of input from non-lemniscal regions of the MGB, such as the MGBm, as well as from the SGN and pulvinar[11,46,47]. Indeed, in mice, the suppressive effects of visual looming stimuli on A1 activity appear to be mediated by the lateral posterior nucleus, the rodent homolog of the primate pulvinar[21]. However, A1 receives the great majority of its ascending input from the MGBv, which is traditionally viewed as a unisensory structure. Nevertheless, cutaneous electrical stimulation has been shown to modulate auditory responses in the MGBv[19,22], and our findings demonstrate that the sound-evoked responses of most neurons recorded there and in the non-lemniscal MGBd are suppressed by concurrent whisker stimulation. Moreover, we observed comparable crossmodal suppression in auditory thalamocortical axon boutons and in A1 neurons, suggesting that somatosensory-auditory interactions are inherited by these cortical neurons from their primary source of thalamic input.

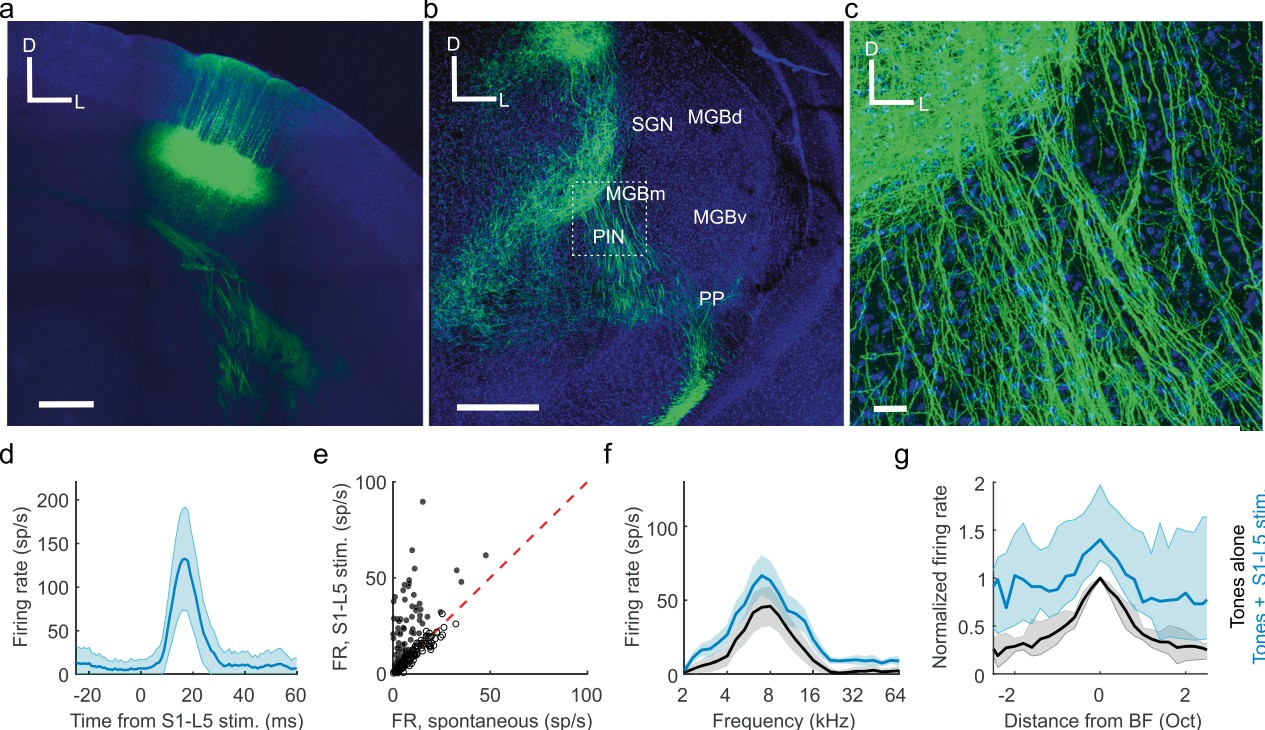

**Fig. 6 Direct pathway from primary somatosensory cortex (S1) to the medial auditory thalamus. a** Confocal image of ChR2-YFP expression in RBP4+ cells in layer 5 (L5) of S1. Scale bar, 400 μm; D, dorsal; L, lateral. **b** Confocal image of a coronal section of the thalamus showing S1-L5 (RBP4+) axons in the medial sector of the auditory thalamus. MGBd, Medial Geniculate Body dorsal division; MGBm, MGB medial division; MGBv, MGB ventral division; PIN, posterior intralaminar nucleus; PP, peripeduncular nucleus; SGN, suprageniculate nucleus. Scale bar, 400 μm. **c** High magnification image (location shown by the dashed box in **b**) showing S1-L5 (RBP4+) axons in MGBm/PIN. Blue = DAPI staining in cell nuclei, Green = YFP in S1-L5 RBP4 + axons. Scale bar, 30 μm. **d** PSTH for an example unit located in MGBm/PIN that was driven by stimulation of S1-layer 5 (RBP4+) neurons. **e** Summary of MGBm/PIN neuronal firing rate (FR) responses to 50 ms light pulses delivered to stimulate S1-L5 (RBP4+) neurons. $n = 183$, 5 mice. Filled circles indicate the 69 units in which spontaneous firing was significantly driven (using a criterion of $P < 0.05$, one-sided $t$ test) by S1-L5 stimulation. **f** Frequency response profiles from an example unit in MGBm/PIN in which the auditory response was significantly enhanced by concurrent stimulation of S1-L5 (RBP4+) neurons. **g** Median frequency response profiles from units in the medial sector of auditory thalamus with significantly ($P < 0.05$, two-sided $t$ test) facilitated BF responses during stimulation of S1-L5 (RBP4+) neurons. $n = 25$, 5 mice. Shaded areas indicate the s.e.m. (**d**), the 95% confidence intervals of the means (**f**) or the 95% nonparametric confidence intervals of the medians (**g**), respectively. BF responses were significantly modulated in 18% (13.7% facilitated, 4.4% suppressed; $n = 183$, 5 mice) of units in MGBm/PIN and SGN by concurrent stimulation of S1-L5 (RBP4+) neurons. Oct, octaves; sp/s, spikes per second.

In the MGBv and MGBd, the strongest suppressive effects induced by whisker stimulation occurred at the BF of the neurons, i.e., the tone frequency at which the largest response was obtained. This crossmodal divisive scaling by non-driving sensory inputs resembles that found in primate cortex[48–50]. The divisive normalization operating in these areas is regarded as a canonical feature of multisensory integration, which can explain the dependence of neuronal responses on the efficacy and spatial relationship of the individual stimuli[49]. Our results suggest that this may be a more widespread property of multisensory neurons, even occurring in a structure (i.e., the auditory thalamus) that lacks recurrent connectivity[51].

In contrast to the exclusively suppressive effects of somatosensory stimulation on the MGBv and MGBd, neurons in the medial sector of the auditory thalamus (MGBm, PIN, and SGN) exhibited a mixture of crossmodal suppression and enhancement and, similar to other species[52,53], ~25% were driven by tactile stimulation. We found that the facilitatory effects of whisker deflection were replicated by optogenetic activation of S1 layer 5 neurons, but were preserved when S1 was silenced, suggesting that they arise from converging corticothalamic and subcortical inputs[54–56]. Neurons in these medial thalamic structures primarily target secondary auditory and higher-level association cortical areas, and the minority that innervate A1 terminate in layer 1 and layer 5/6[31,57,58]. However, the thalamic axon boutons that we imaged in layer 1 showed exclusively crossmodal suppression of sound-evoked activity, suggesting that neurons whose responses are facilitated by somatosensory inputs likely project elsewhere. Non-cortical targets of the medial auditory thalamus include the basal ganglia[31,59] and amygdala[31,53,57,60], with the latter projection being a critical part of the circuitry mediating auditory fear conditioning[60–62].

In addition to differences in their efferent targets and in the effects of somatosensory inputs on their responses to sound, the physiological properties of neurons in the MGBm, PIN, and SGN are distinct in other ways from those in the MGBv/MGBd[63]. Indeed, the lack of excitatory connectivity between these neurons[51] makes the auditory thalamus an ideal place to establish functionally distinct pathways that are independently and flexibly modulated by contextual information, including inputs from other senses or motor commands[24].

**Corticofugal crossmodal control of the auditory thalamus.** Descending corticofugal pathways play a critical role in processing sensory information, both within and across sensory modalities, and in integrating sensory and motor signals[24,33,34,64,65]. Auditory cortical feedback can inhibit MGB activity via

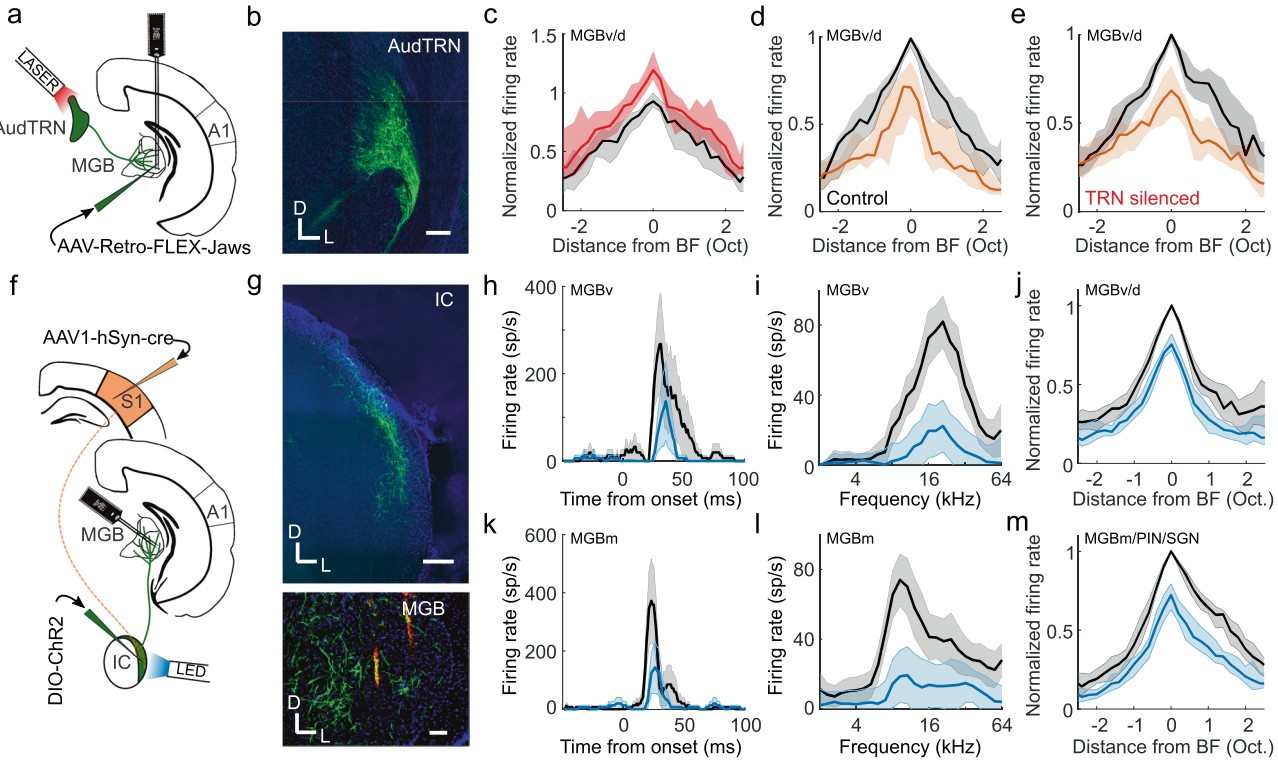

**Fig. 7 Corticocollicular circuit mediates somatosensory suppression of the thalamus. a** Schematic of experimental paradigm in **b**–**e**. A1, primary auditory cortex; AudTRN, auditory sector of the thalamic reticular nucleus; MGB, medial geniculate body. **b** GABAergic cells in TRN retrogradely-labeled with Jaws from auditory thalamus. Scale bar, 150 μm. **c** Summary (median) frequency tuning curves across MGBv/d units with (red) or without (black) optogenetic suppression of AudTRN activity (change in BF firing response, $P = 1.1 \times 10^{-6}$, $n = 38$, 2 mice, two-sided Wilcoxon signed-rank test). **d**, **e** Median frequency response profile of MGBv/MGBd units (same units as in **c**) illustrating suppression induced by concurrent whisker stimulation (orange) with AudTRN either unaffected (**d**) or optogenetically silenced (**e**). Silencing AudTRN had no overall effect on the whisker-induced suppression of auditory responses in MGBv/MGBd ($P = 0.83$, $n = 38$, 2 mice, two-sided Wilcoxon signed-rank test) and there was no relationship between the change in auditory response magnitude and the effect on whisker-driven suppression of the auditory response (Pearson's $r = -0.055$, $P = 0.74$). **f** Schematic of experimental paradigm in **g**–**m**. **g** Top: ChR2-YFP expression in neurons in the shell of IC, labeled by anterograde transport of cre from S1 (AAV1-hSyn-cre) and a cre-dependent AAV5-DIO-ChR2-eYFP injected into the IC. Scale bar, 200 μm. Bottom: Axons (green) of anterogradely labeled IC neurons in MGB. Scale bar, 100 μm. Orange marks show DiI tracts from the recording probe in the MGB. D, dorsal; L, lateral. **h** Example PSTHs illustrating BF responses of an MGBv unit with (blue) and without (black) optogenetic stimulation of S1-recipient IC neurons. **i** Example frequency response profile of an MGBv unit with (blue) and without (black) optogenetic stimulation of S1-recipient IC neurons. **j** Median MGBv/MGBd frequency response profile with (blue) and without (black) stimulation of S1-recipient IC neurons: −20.9% median change in BF firing rate ($P = 1.4 \times 10^{-14}$, $n = 85$, 3 mice, two-sided Wilcoxon signed-rank test). **k**–**m** same as **h**–**j** for units recorded in MGBm/PIN/SGN. **m** −26.9% median change in BF firing rate ($P = 3.5 \times 10^{-14}$; $n = 89$, 3 mice, two-sided Wilcoxon signed-rank test). Shaded area illustrates the s.e.m. (**h**, **k**), the 95% confidence intervals of the means (**i**, **l**), or the 95% nonparametric confidence intervals of the median (**c**, **d**, **e**, **j**, **m**). MGBm/PIN/SGN, MGB medial division/posterior intralaminar nucleus/suprageniculate nucleus; MGBv/d, MGB ventral/dorsal divisions; sp/s, spikes per second.

GABAergic neurons in the TRN[66], but this pathway does not appear to be responsible for somatosensory suppression of auditory thalamic responses. Instead, we have identified a descending projection from S1 to IC shell neurons that can inhibit responses in the MGB. Somatosensory dominance over auditory processing in mouse A1, therefore, appears to be implemented by a corticocolliculo–thalamocortical circuit. These findings add to the growing evidence that trans-thalamic circuits enable communication between cortical areas[23], and demonstrate that the midbrain is also part of the circuitry responsible for integrating multisensory signals across the cerebral cortex.

Interactions between somatosensory and auditory inputs occur as early as the cochlear nucleus in the brainstem[67]. We did not observe any effects of whisker stimulation on the auditory responses of neurons recorded in the CNIC, the primary relay nucleus of the auditory midbrain, suggesting that multisensory suppression in the MGBv is unlikely to be inherited from earlier in the auditory pathway. In contrast, somatosensory-auditory interactions are prevalent in the IC shell. The LCIC is of

particular interest since it receives inputs from much of the body surface via projections from the somatosensory cortex and the brainstem[37,68]. In mice, these inputs target GAD-67-positive modules that are separated by regions receiving auditory inputs[37]. Furthermore, GABAergic neurons throughout the IC project to the MGB[38,39,69,70]. Our findings bridge these studies and establish a functional role for such circuits by demonstrating that a relatively small population of GABAergic S1-recipient neurons in the lateral shell of the IC can account for the suppressive effects of whisker stimulation on sound-evoked responses in the auditory thalamocortical system.

**Perceptual implications of somatosensory control over auditory processing.** Given its key position in the brain, context-dependent modulation of neuronal activity in the thalamus has wide-ranging consequences for information processing, not only in the cerebral cortex but also in other thalamorecipient brain regions, such as the amygdala and basal ganglia. The presence of region-specific multisensory interactions throughout the auditory

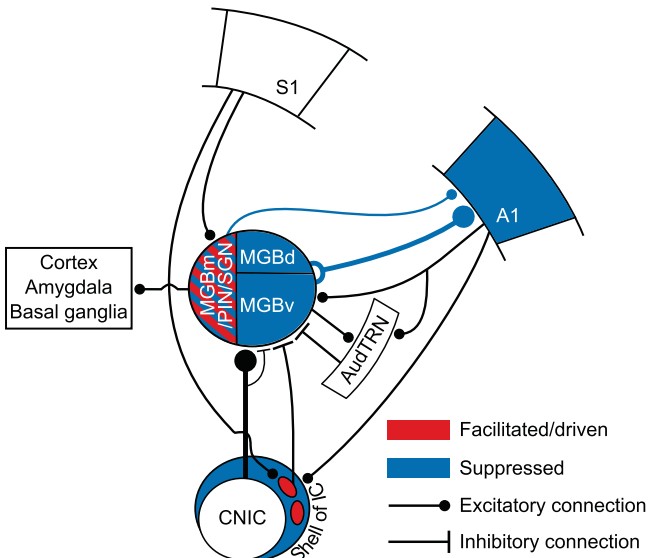

**Fig. 8 Circuits enabling somatosensory control of the auditory thalamocortical system.** Auditory responses in the regions of the auditory thalamus and cortex depicted in blue were suppressed by concurrent whisker stimulation via a descending pathway from S1 to the lateral shell of IC, which then projects to the MGB. Some neurons in the medial sector of the auditory thalamus were driven or had their auditory responses enhanced by whisker stimulation (depicted in red), which can be mediated by a direct corticothalamic projection from S1 to MGBm/PIN/SGN. A1, primary auditory cortex; AudTRN, auditory sector of the thalamic reticular nucleus; CNIC, central nucleus of the inferior colliculus; MGBm/PIN/SGN, MGB medial division/posterior intralaminar nucleus/suprageniculate nucleus; MGBd, MGB dorsal division; MGBv, MGB ventral division; S1, primary somatosensory cortex.

thalamus, therefore, implies that combining information from different sensory modalities at this relatively early stage in the processing hierarchy plays a fundamental role in how animals perceive and interact with their sensory environments.

In rats, facial touch is associated with inhibition of auditory cortical activity[6], potentially reflecting a greater salience of haptic information during social interactions and exploration. Our data suggest that these effects are present in the thalamus too and that they are asymmetric since we observed a much weaker modulatory influence of sound on neuronal responses to whisker stimulation in the somatosensory thalamus and no effect on whisker responses in the S1 barrel field (Supplementary Fig. 14). Suppressive effects of somatosensory stimulation on sound-evoked responses are also thought to reduce the impact of vocalizations or other self-generated and potentially distracting sounds, such as those resulting from chewing or breathing[20].

Although somatosensory suppression of auditory thalamocortical activity may reflect the relative importance of these inputs when nearby objects are encountered during exploration of the environment, a reduction in the firing rate of auditory neurons in the presence of other sensory cues can be accompanied by an increase in response reliability and in the amount of stimulus-related information transmitted[5,71]. Furthermore, auditory cortical activity is suppressed when an animal engages in a task[72]. Of particular relevance to the present study is the finding that divisive scaling of auditory cortical frequency tuning, as demonstrated in our recordings, is associated with improved frequency discrimination at the expense of impaired tone detection[65]. By inducing divisive gain changes in the auditory thalamocortical system, somatosensory inputs might function as a

bottom-up cue that sharpens auditory acuity, whilst reducing sensitivity.

## Methods

**Mice.** All experiments were approved by the Committee on Animal Care and Ethical Review at the University of Oxford and were licensed by the UK Home Office (Animal Scientific Procedures Act, 1986, amended in 2012). Seven strains of male and female mice were used: C57BL6/J (Envigo, UK), VGAT-ChR2-YFP (JAX 014548—Jackson Laboratories, USA), VGAT-cre (JAX 016962—Jackson Laboratories, USA), Ai95 (RCL-GCaMP6f)-D (JAX 024105—Jackson Laboratories, USA), Ai95 (RCL-GCaMP6f)-D (JAX 024105—Jackson Laboratories, USA), X VGAT-cre (JAX 016962—Jackson Laboratories, USA), Ai9 (RCL-tdT) (JAX 007909—Jackson Laboratories, USA) and C57BL6/NTac.Cdh23 (MRC Harwell, UK). C57BL6/NTac.Cdh23 mice[73] were 10–20 weeks old; all others were 7–12 weeks old at the time of data collection. Mice were maintained on a 12-h light/dark cycle and were housed at 20–24 °C with a relative humidity of 45–65%. All experiments were carried out in sound-attenuated chambers.

**Stimuli.** Auditory stimuli were programmed and controlled in custom-written Matlab code (https://github.com/beniamino38/benware) and generated via TDT RX6 (electrophysiology) or RZ6 (2-photon imaging) microprocessors (Tucker-Davis Technologies, USA). Sounds were generated at a ~200 kHz sampling rate, amplified by a TDT SA1 stereo amplifier, and delivered via a modified (i.e., sound was "funnelled" into an otoscope speculum) Avisoft ultrasonic electrostatic loudspeaker (Vifa, Denmark, for the electrophysiology) or a TDT EC1 electrostatic speaker (imaging) positioned ~1 mm from the entrance to the ear canal. The sound presentation system was calibrated to a flat (±1 dB) frequency-level response between 1 and 64 kHz. Stimuli included pure tones, covering a frequency range from 2 to 64 kHz, and broadband noise bursts (1–64 kHz). All sounds included 5-ms linear amplitude onset/offset ramps, and unless specified otherwise were presented at 80 dB SPL.

Whisker deflections were delivered with a piezoelectric bimorph attached to a small glass tube. During stimulation, the majority of the whiskers were either positioned inside the stimulation tube (anesthetized recordings), or a small brush with plastic hairs was attached to the tube in which whiskers were interspersed in the hairs of the brush (awake recordings). We deflected the whiskers in a single cosine wave (valley-to-valley), transiently displacing the whiskers 1 mm from resting position at a speed of 40 mm/s.

Presentation of acoustic and whisker stimuli (or both together) was randomly interleaved, with each sensory stimulus having a duration of 50 ms, unless otherwise specified.

**Extracellular recordings.** We carried out extracellular recordings using 32- or 64-channel silicon probes (NeuroNexus Technologies Inc.) in a 4 × 8, 8 × 8 or 2 × 32 electrode configuration. Prior to insertion, probes were coated with DiI (Sigma-Aldrich) for subsequent histological verification of the recording sites. Data were acquired using a RZ2 BioAmp processor (TDT) and custom-written Matlab code (https://github.com/beniamino38/benware).

For recordings under anesthesia, mice were anesthetized with an intraperitoneal (ip) injection of ketamine (100 mg kg$^{-1}$) and medetomidine (0.14 mg kg$^{-1}$). Atropine (Atrocare ip, 1 mg kg$^{-1}$) to prevent bradycardia and reduce bronchial secretions and dexamethasone (Dexadreson ip, 4 mg kg$^{-1}$) to prevent brain edema were administered. Prior to the surgery, the analgesic bupivacaine was injected under the scalp. The depth of anesthesia was monitored via the pedal reflex and adjusted with small additional doses of the ketamine/medetomidine mix (1/5th of the initial dose) given subcutaneously approximately every 15 min once the recordings had started (~1–1.5 h post-induction of anesthesia). A silver reference wire was positioned in the visual cortex of the contralateral hemisphere and a grounding wire was attached under the skin on the neck musculature. The head was fixed in position with a metal bar attached to the skull with dental adhesive (Super Bond C&B).

For awake recordings in the auditory thalamus and auditory cortex, we implanted a recording chamber under isoflurane (1.5–2% in O$_2$) general anesthesia. Mice received ip injections of buprenorphine (Vetergesic 1 ml/kg), dexamethasone (Dexadreson 4 μg), and atropine (Atrocare 1 μg). An additional dose of buprenorphine was given 24 hours post-operatively. The recording chamber consisted of a well that was constructed out of dental adhesive (Super Bond C&B) encircling the craniotomy, which was sealed with a circular glass window. We positioned the recording chamber either above the visual cortex (centered ~3 mm caudal from bregma and ~2.1 mm lateral from midline) for auditory thalamus recordings, or above A1 (centered ~2.5 mm posterior from bregma and ~4.5 mm lateral from midline), together with a head bar, and placed a reference electrode (silver wire) in the contralateral hemisphere. One or two days later the mouse was head-fixed, the recording chamber opened, and a sterile recording probe acutely inserted into the brain via the recording chamber.

All recordings were performed in the right hemisphere. In the anesthetized preparation, circular craniotomies (2 mm diameter) were performed above the IC (centered ~5 mm posterior from bregma and ~1 mm lateral from midline), over

the visual cortex for auditory thalamic recordings and/or over A1. The exposed dura mater was kept moist with saline throughout the experiment.

Recording sites in the different subdivisions of the IC were confirmed by post-mortem brain histology. In addition, recording sites were considered to be in the CNIC when the units recorded on those sites were part of a clear dorso-ventral tonotopic gradient[74,75]. For recordings in the MGB, probe sites were attributed to specific auditory thalamic subdivisions by histological reconstruction of the recording sites (Supplementary Fig. 2). We parcellated the auditory thalamus based on previous immunohistochemical descriptions[29] and our own pilot tracing experiments from several cortical areas (including from S1 and A1). Accordingly, recording sites were assigned to the ventral division (MGBv), dorsal division (MGBd), medial division and posterior intralaminar nucleus (MGBm/PIN), or SGN. Based on these histological reconstructions, recording sites attributed to the MGBv were located <500 μm from the lateral border of the MGB and <500 μm from the deepest acoustically-responsive site, while those in the MGBd were <500 μm from the lateral border of the MGB, but >500 μm from the most ventral acoustically-responsive site. For recordings in the medial sector of the auditory thalamus, sites assigned to the MGBm/PIN were >500 μm from the lateral border of the MGB and <500 μm from the most ventral acoustically-responsive site, and those in the SGN were >500 μm from the lateral border of the MGB and >500 μm from the most ventral acoustically-responsive site.

A1 was identified by robust neuronal responses to broadband noise bursts, well-tuned neurons, and a well-defined caudo-rostral tonotopic axis[31,76]. Cortical tonotopy was assessed in all anesthetized cortical recordings by estimating frequency response areas from responses to pure tones using probes with four recording shanks spaced 200 μm apart and oriented parallel to the caudo-rostral axis. Recordings in awake animals were performed in positions corresponding to those identified as A1 from the anesthetized cases.

### Two-photon calcium imaging

*Imaging thalamocortical axons and boutons in primary auditory cortex.* All viral vector injections were performed using a custom-made pressure injection system with a calibrated glass pipette positioned in the right hemisphere. We made injections of ~140 nl (diluted 1:1 in PBS) of AAV1.Syn.GCaMP6m.WPRE.SV40 into the auditory thalamus (3 mm caudal from bregma, 2.1 mm lateral from midline and 2.8–3 mm ventral from the cortical surface) for expression of GCaMP6m in auditory thalamic neurons and axons as reported previously[31]. In order to visualize the calcium activity of thalamic boutons in layer 1 (20–80 μm below the surface) of the auditory cortex, mice were chronically implanted with a head bar and a circular 4 mm diameter glass window. The implant surgery procedure took place 2–3 weeks following injection of the viral construct. All the viral vector injections and implants were performed under Isoflurane (1.5-2% in O2) under general anesthesia. Data acquisition began ~7 days after the implant surgery. As with the extracellular recordings under anesthesia, mice were kept anesthetized with a mixture of ketamine and medetomidine throughout the experiment.

*Imaging GABAergic neurons in primary auditory cortex.* Expression of GCaMP6f was targeted to GABAergic neurons by crossing Ai95 (RCL-GCaMP6f)-D (JAX 024105—Jackson Laboratories, USA) with VGAT-cre (JAX 016962—Jackson Laboratories, USA) mice. The mice were fitted with identical implants and cranial windows as described above. Data were obtained from neurons in layers 2/3 (150–250 μm below the surface) and while the animals were awake. A1 was localized using widefield imaging as described previously[77].

*Imaging neurons in the dorsal cortex of the inferior colliculus.* Expression of GCaMP6f was targeted to IC shell neurons by injecting ~140 nl of the trans-synaptically transported AAV1-hSyn-cre into the auditory cortex of Ai95 (RCL-GCaMP6f)-D (JAX 024105—Jackson Laboratories, USA) mice. The mice were fitted with implants for head fixation and a circular glass window (3 mm diameter) was inserted over the IC. Data were obtained while the animals were awake and from neurons just beneath the dorsal surface of the IC (50–150 μm below the surface).

All calcium imaging was carried out using a 2-photon laser scanning microscope (B-Scope, Thorlabs, USA). Excitation light of 930 nm (10–50 mW power measured under the objective) was provided by a Mai-Tai eHP (Spectra-Physics, USA) laser fitted with a DeepSee prechirp unit (70 fs pulse width, 80 MHz repetition rate). The laser beam was directed through a Conoptics (CT, USA) modulator and scanned onto the brain with an 8 kHz resonant scanner (x-axis) and a galvanometric scan mirror (y-axis), allowing acquisition of 512 × 512 pixel frames at ~30 Hz. Emitted photons were guided through a 525/50 filter onto GaAsP photomultipliers (Hamamatsu, Japan). We used ScanImage[78] to control the microscope during data acquisition and a 16× immersion objective (Nikon, Japan).

### Viral injections and transgenic expression of proteins for optogenetic control.

The tip of the pipette was carefully and slowly inserted into the area of interest, and ~20 nl boluses were then given every two minutes until the desired volume had been injected. The pipette was then left in position for an additional 5 min before being slowly retracted. All optogenetic experiments involving viral injections were carried out >3 weeks after the injection to allow for expression of the opsin. All optogenetic stimulation experiments were carried out with a bright white LED shining into the eyes of the mouse throughout the experiment, to saturate photoreceptor responses in the retina and prevent visual activity being induced by the light stimulation[79].

*Activating infragranular cells in S1 using ChrimsonR whilst imaging auditory thalamocortical axons and boutons in A1.* We injected 120 nl of AAV1-CAG-ChrimsonR[80] in S1 (−0.8 and −1.0 mm caudal from bregma, 2.6 mm lateral from midline, and 0.8, 0.65, and 0.5 mm ventral from the cortical surface) to induce expression in the infragranular layers of S1 of C57BL6/J mice. In the same surgery, we also injected AAV1-hSyn-GCaMP6m into auditory thalamus and implanted a glass window over the auditory cortex and a head bar, as explained previously. Finally, in the same surgery, we placed a 400 μm fiber optic cannula on the dura above S1. For optogenetic activation, a 3 mW, 595 nm LED pulse (Doric Lenses, Canada) was delivered to S1 concurrently with, and for the duration of, broadband noise stimulation (i.e. 50 ms or 200 ms).

*Activating RBP4 + cells in layer 5 of S1 using ChR2.* We injected 60–80 nl of AAV5-DIO-hChR2-eYFP[81] in S1 (using the same rostrocaudal and mediolateral coordinates as in the previous experiment, and 1.0, 0.95 and 0.9 mm ventral from the cortical surface) of RBP4-cre mice to induce expression of ChR2 in layer 5 neurons. For optogenetic activation, a 20 mW, 465 nm LED pulse (Doric Lenses) was presented. Light was delivered through a 1 mm fiber acutely positioned on the dura mater above S1 concurrently with sound stimulation (i.e., for a duration of 50 ms).

*Suppressing neuronal activity in the auditory sector of thalamic reticular nucleus using Jaws.* In order to transfect cells in the auditory sector of TRN (audTRN) with Jaws, we exploited the fact that the MGB in rodents contains very few inhibitory cells[35]. An injection of 140 nl of the cre-dependent retrograde construct pAAV-CAG-FLEX-rc[Jaws-KGC-GFP-ER2][82,83] was placed into the MGB of VGAT-cre mice. The construct did not label cells inside the MGB, but instead induced Jaws expression in cre-expressing TRN cells that project to the injection site in the auditory thalamus. After the injection, we placed a 400 μm fiber optic cannula immediately above audTRN. To maximize the light transmission to the transfected area of audTRN the fiber optic cannula was implanted at a 22.5° angle (relative to the coronal axis). The anatomical position was histologically confirmed after the end of the experiments. For optogenetic suppression, we used a 60 mW, 640 nm laser pulse (Toptica Photonics) of 150 ms length, which started 25 ms before sound onset.

*Intersectional targeting and activation of S1-recipient neurons in the shell of the IC.* We induced expression of cre in neurons receiving projections from S1, by injecting 200 nL of AA1-hSyn-cre into S1 (at 0.9, 0.7, and 0.5 mm ventral from the cortical surface). This virus anterogradely and trans-synaptically infected neurons receiving projections from S1 and induced expression of cre in those neurons[40]. In order to target expression of ChR2-YFP to IC neurons that receive input from S1, we also injected 200 nL of the cre-dependent construct AAV5-DIO-ChR2-YFP into the lateral part of the IC. For optogenetic activation, a 20 mW, 465 nm LED pulse (Doric Lenses) was delivered through a 1 mm optic fiber acutely positioned on the dura mater above the lateral part of the dorsal IC. Stimulation occurred concurrently with sound stimulation (i.e., for a duration of 50 ms).

*Silencing excitatory cortical activity in VGAT-ChR2-YFP mice.* For optogenetic silencing of A1 and S1, we used a blue (465 nm) LED stimulus (duration 150 ms, onset 25 ms before auditory and/or somatosensory stimulation) delivered via a 200 μm optic fiber (Doric Lenses) acutely implanted over the dura mater above A1 or the S1 barrel field, respectively. ChR2 was targeted to GABA neurons using VGAT-ChR2-YFP mice. Light power was 2.5 mW.

*Identifying GABAergic IC neurons that receive input from S1.* VGAT-YFP-ChR2 mice were used to achieve double labeling of GABAergic IC neurons that receive input from S1. They received injections of ~140 nl of AAV1-hSyn-cre into S1 plus ~140 nl of AAV1-CAG-Flex-tdTomato-WPRE-bGH into the lateral part of IC.

### Histology.

For post-mortem verification of the electrophysiological recording sites, viral expression pattern, and anatomical tracing, mice were overdosed with pentobarbital (100 mg/kg body weight, i.p.; pentobarbitone sodium; Merial Animal Health Ltd, Harlow, UK) and perfused transcardially, first with 0.1 M phosphate-buffered saline (PBS, pH 7.4) and then with fresh 4% paraformaldehyde (PFA, weight/volume) in PBS. Mice used in anatomical experiments were euthanized and perfused >4 weeks after the virus injections. Mice used for electrophysiology were perfused as soon as the recordings were finished (acute experiments) or when the last recording session was finished (awake recordings), while those used for chronic 2-photon imaging were perfused when all imaging sessions were completed. Following perfusion, the brain was removed from the skull and kept in 4% PFA (weight/volume) in PBS for ~24 h. The relevant parts of the brains were then sectioned using a vibratome in the coronal plane at a thickness of 50 or 100 μm. Sections were mounted on glass slides and covered in a mounting medium with DAPI (Vectashield, Vector Laboratories). Images were acquired with an Olympus

FV1000 confocal laser scanning biological microscope. Confocal images were captured using similar parameters of laser power, gain, pinhole, and wavelengths with up to three (red, green, blue) channels assigned as the emission color; z-stacks were taken individually for each channel and then collapsed. Images were processed offline using Imaris (Zurich, Switzerland) and ImageJ (NIH, MD, USA).

**Data analysis and statistics.** We clustered potential neuronal spikes using KiloSort[84] (https://github.com/cortex-lab/KiloSort). Following this automatic clustering step, we manually inspected the clusters in Phy (https://github.com/kwikteam/phy) and removed noise (movement and optogenetic light artifacts). We assessed clusters according to suggested guidelines published by Stephen Lenzi and Nick Steinmetz (https://phy-contrib.readthedocs.io/en/latest/template-gui/#user-guide). Each cluster (following merging and noise removal) was assigned as either noise (clearly not neuronal spike shape), multi-unit (neuronal and mostly consistent spike shape with no absolute refractory period), or single unit (consistent spike shape with absolute refractory period). All analyses performed on the electrophysiological data were run on a combination of small multi-unit clusters and single units (no differences were found between them and therefore we just refer to these as units). Stimulus-evoked responses were measured as the mean firing rate (spikes per second, sp/s) for the duration of the stimulus presentation. Baseline activity was measured from the mean firing rate of the 90 ms preceding stimulus onset.

For 2-photon imaging of thalamocortical axons and boutons, we carried out standard preprocessing (e.g., registration of image stacks, region of interest selection, trace extraction) of the calcium data, as described in detail elsewhere[31,41]. Given the slower dynamics of GCaMP6m used to monitor bouton activity from auditory thalamocortical axons, we measured the calcium transient response to a 50 ms stimulus as the mean $\Delta F/F$ over the 16 frames following stimulus onset (i.e., for ~550 ms). Baseline activity was measured as the mean $\Delta F/F$ over the 16 frames preceding stimulus onset. For preprocessing of cell body calcium imaging data and spike detection, we used Suite2p[85] and the OASIS deconvolution algorithm[86].

For estimation of somatosensory modulation of noise responses, we only included units/boutons that showed a statistically significant response during sensory stimulation compared to baseline (one-sided $t$ test, $P < 0.005$). For estimation of somatosensory modulation of tone responses, we only included units/boutons that showed a statistically significant difference in response among the frequency-level combinations tested (one-way ANOVA, $P < 0.005$).

The BF of tone-responsive neurons and boutons was defined as the sound frequency associated with the largest response (i.e., firing rate or $\Delta F/F$, respectively) at the sound level used. For summary statistics and display of summary frequency tuning across units/boutons, we normalized the frequency response profile of each unit/bouton. To do this, we first estimated the mean frequency response profile across conditions (e.g., with and without whisker deflection and/or S1/A1 manipulations), and centered the response profiles for each condition on the BF estimated from the mean response profile. We then normalized the response to each tone frequency presented—separately for each condition—by dividing by the response at the BF in the control condition (i.e., tones presented alone). We then produced a summary frequency response profile by taking the median of the normalized response profile across units/boutons. Error bars for the summary response profiles were estimated from bootstrapped (10,000 iterations) 95% nonparametric confidence intervals.

For group (i.e., across units or boutons) comparisons, we used nonparametric statistical tests (i.e., Wilcoxon signed-rank for paired samples and Mann–Whitney $U$ test for independent samples).

Example images showing viral expression patterns were reproduced in all animals using similar injections for circuit manipulation during recording/imaging.

**Reporting summary**. Further information on research design is available in the Nature Research Reporting Summary linked to this article.

## Data availability
All relevant data are available on request, and will be fulfilled by, the lead contact (michael.lohse@dpag.ox.ac.uk). Raw electrophysiological recording and imaging data files are too large to be placed in an online repository, but are available upon reasonable request via the lead contact (michael.lohse@dpag.ox.ac.uk). Source data are provided with this paper and data demonstrating the analyses used are available on GitHub: https://github.com/LohseNeuro/Somato_Auditory_Circuits-Lohse-et-al-2021 (https://doi.org/10.5281/zenodo.4790471).

## Code availability
Matlab code for analyses are available on request to, and will be fulfilled by, the lead contact (michael.lohse@dpag.ox.ac.uk). Scripts used for analyses are available on GitHub: https://github.com/LohseNeuro/Somato_Auditory_Circuits-Lohse-et-al-2021 (https://doi.org/10.5281/zenodo.4790471).

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

## Acknowledgements

The research was funded by a Wellcome Trust Studentship (WT105241/Z/14/Z) to M.L., and a Wellcome Trust Principal Research Fellowship (WT108369/Z/2015/Z) to A.J.K. We thank Christopher Breen for helping with the histology.

## Author contributions

M.L. conceived the study. M.L, J.C.D., and A.J.K. designed the experiments. M.L. and J.C.D. performed the research. M.L. analyzed the data. M.L. and A.J.K. acquired funding for the research. A.J.K. provided infrastructure and resources. A.J.K, J.C.D., and V.M.B. supervised the research. M.L., J.C.D., V.M.B., and A.J.K. interpreted the research. M.L., J.C.D., V.M.B., and A.J.K wrote the manuscript.

## Competing interests

The authors declare no competing interests.
