## [Peer Review File · Nature Communications]

Reviewer #1 (Remarks to the Author):

Lohse and colleagues describe new subcortical pathways by which whisker stimulation suppresses tone-induced neuronal responses in auditory thalamus and primary auditory cortex. The authors show that whisker stimulation takes effect through primary somatosensory cortex via the auditory midbrain and auditory thalamus.

This is an interesting and impressive study with an incredible amount of data. I have only few comments, most of them minor.

1) My main criticism is that the authors do not show directly that the pathway through the IC shell is conveying whisker-induced suppression of tone responses in auditory thalamus and cortex. They instead show that stimulating IC cells receiving S1 input can suppress thalamic responses. It would strengthen the conclusions if the authors could silence the IC shell during tone and whisker stimulation and recordings in auditory thalamus (and cortex) similar to the experiments in Figure 5a-c for S1 silencing. However, these experiments are very challenging and if the authors are not able to perform them, in my opinion the study is strong enough without these data, but the authors should mention this pitfall in the discussion.

2) Related to point 1, the authors do not show that direct connections from S1 to A1 do not convey at least part of the whisker-induced response suppression in A1. Again, experiments providing conclusive evidence for or against a contribution of direct cortical connections are very challenging and the study provides enough important new findings without these data, but the authors should discuss this more directly in the discussion section.

Minor:

3) It would be good to include some of the important numbers and stats in the manuscript text.

4) Line 152: the title "Somatosensory suppression of auditory cortex is inherited from the thalamus" is misleading. It is of course likely that this is the case since vMGB provides a major drive to A1, but the authors do not show this directly, only that thalamic neurons projecting to layer 1 of A1 show suppression of responses by whisker stimulation. The authors should change the section title accordingly.

5) Figure 3: is there a difference in the distribution of effects of whisker stimulation in the different medial subsections of the auditory thalamus? Also please indicate the location of the map in Figure 3j in the thalamic.

6) Suppl. Figure 7: I do not see a decrease in the whisker-induced response suppression when silencing S1 in Suppl. Figure 7b.

7) Do RBP4-positive neurons also project to IC, or a different population of layer 5 cells?

8) Figure 6c: please add information on the location of this image, e.g. by adding a square in Figure 6b.

9) Please add clearer images in Figure 7g to show cells in IC and location and density of axons in MGB.

Reviewer #2 (Remarks to the Author):

In this manuscript, Lohse and colleagues report on the influence of the somatosensory system on auditory responses in midbrain, thalamus and cortex to study the details of integration of information across senses. This is an important addition to the literature as it is of paramount importance to shed light on our general understanding on perception.

Authors show that stimulation of the whiskers causes widespread suppression of sound-evoked activity in mouse primary auditory cortex. Authors further demonstrate that this suppression depends on the primary somatosensory cortex and is implemented through a descending circuit that links S1, via the auditory midbrain, with thalamic neurons that project to A1.

Overall I like the manuscript. The study is multidisciplinary and use several complementary techniques It is clearly written and easy to follow, so I have no major issues. Perhaps the section on the discussion is a bit too long and authors could make an effort to trim down a bit so it is more focused and distinctive. I particularly like the subcortical emphasis that authors make as opposed to many other studies on cortex that tend to be corticocentric and to neglect the subcortical contribution to sensory processing. I also like the combination of awake and anaesthetized preparations. Again, I like the approach where authors give more importance to the anaesthetise, this is an appeal to the overall demand of awake preparations by many journals (including Nat Comm, as I have suffered of it myself). This is not to say I dislike awake, simply that it is not a mandatory experiment. Anyway.

So in general I am supportive for the study to be published in Nat Comm but I also have a few comments that authors may care to consider, or at least comment on it in the paper or rebuttal letter.

Specific comments

major

An important question, at least to me is that whilst authors have recorded from the CNIC and found no somatosensory-auditory interactions, this is not surprising at all. But I wonder what would happen if similar recordings are done in the non-lemniscal IC, i.e., in the lateral, dorsal and rostral regions surrounding the IC that are well known (as the authors themselves cite) to receive somatosensory inputs. This possibility needs to be further explored, ideally with additional experiments or at the very least, this limitation should be discussed.

Another important issue is the data from MGB and particularly that, shown in supplementary fig 4. Authors state: 'Similar somatosensory influences on auditory responses were found in MGBv/d of awake, head-fixed mice (Supplementary Fig. 4).' However, I find that although statistically significant it is very marginally and wonder the functional significance of this strong statement. In connection with this issue, authors later on study the medial part of the MGB. So my big question here is that, either I have misunderstood, or authors have mix up lemniscal and non lemniscal MGB. The MGBv/d is analysed as a pool but in my humble view this is inappropriate. It would-be more reasonable to pool MGBd and MGBm, PP and PIL.... As these are the genuine non lemniscal regions. Then, perhaps the statistics are different, more robust and clear. The fact that responses in the medial MGB regions are more heterogenous does not mean are less important or functionally significant, and this should be further explored and discussed.

Minor

The first sentence of the results could go to the end of intro, or at least it would be very informative to state right at the end of Intro what are you going to do. I miss a specific aim. I think this would be very helpful for a general audience.

Although authors refer to Figure 1 b and c. as tuning curves, strictly speaking they are not so, I would

rather refer to them as isointensity functions that reflect the BF.

I wonder if the panel from fig 2a-c could be combined with wig 1, so the reader can directly compare the AC and MGB,

Also, I think it would be very valuable if authors include the MGB borders in the Voronoi diagram from fig 3j.

I encourage the authors to engage in scientific discussion with me and other reviewers, nothing of what I say here is a must, I just try to be critical but constructive take on this lovely work you have done

Reviewer #3 (Remarks to the Author):

In this manuscript, Lohse et al. first describe a divisive normalization process in the auditory cortex (AC) when sound and whisker stimuli are presented simultaneously. They then show that this multisensory modulation occurs in the dorsal and ventral MGB (dMGB, vMGB) and not in the central nucleus of IC (CNIC). In contrast to the dMGB and vMGB, mMGB neurons show a mix of facilitation and inhibition by touch. The authors go on to show that the diminished responses in dMGB and mMGB are not driven by the auditory portion of the thalamic reticular nucleus (TRN). Finally, they use trans-synaptic activation of presumed inhibitory neurons from shell regions of IC and show that activation of these neurons mimics the same suppression seen by multimodal stimulation.

This work makes an important contribution to our understanding of multimodal integration. Indeed, it postulates a very novel hypothesis that long-range somatosensory projections to the shell IC are important for suppressive auditory-somatosensory interactions, and provides evidence to support this hypothesis. The manuscript is well-written and illustrated. There are some weaknesses that should be addressed:

1. Do the RBP4+ cells in layer 5 of somatosensory cortex not project to the IC? Given mounting data that suggest that layer 5 cortico-thalamic cells branch to the midbrain, one would assume that these neurons also project to IC. If so, one would expect that their activation would engage the inhibitory neurons in the shell IC, as postulated in Fig 8, and suppress multimodal responses in the MGB (and AC). Please comment on this.
2. The authors speculate that inhibitory neurons from the shell IC targeted by the somatosensory cortex provide ascending inhibition to the MGB. Another possibility is that shell IC neurons (excitatory or inhibitory) could engage local circuits at the level of the IC that then modulate the inhibitory ascending projections to the MGB. To that end, it would have been interesting to identify the target IC neurons as being excitatory or inhibitory and the degree to which they project locally. The authors should modify their model in Fig 8 to account for the potential that somatosensory cortex engages local circuits that then send a projection to the MGB. Also, why are no cell bodies seen in the image in 7G?
3. Were the TRN data obtained in awake animals? If not, the authors should comment that modulating the TRN in an anesthetized animal is unlikely to account for its role in multisensory processing given the strong modulation of this nucleus by arousal. In addition, the authors should specify that the two animals showing the mild effects of TRN stimulation (fig 7C) were the same animals in Figs 7e and f. Also, was there any relationship between the degree of modulation that any one cell had in terms of TRN general effects vs. TRN multimodal effects? If there is a linear

relationship then the negative results may be explained by a large subset of cells whose TRN inputs were not adequately silenced.

4. Imaging thalamocortical terminals in layer 1 will miss the majority of thalamocortical terminals. Therefore, it is not appropriate to conclude that multimodal stimulatory signals are not relayed to the cortex. Therefore, Fig 4 adds very little to the manuscript. It is, however, appropriate to keep the thalamocortical terminal imaging in Fig 5.

Minor:

1. There are two references on auditory-somatosensory suppressive interactions that they authors may consider including:

Dehner, L. R., Keniston, L. P., Clemo, H. R. & Meredith, M. A. Cross-modal Circuitry Between Auditory and Somatosensory Areas of the Cat Anterior Ectosylvian Sulcal Cortex: A 'New' Inhibitory Form of Multisensory Convergence. *Cerebral Cortex* 14, 387-403, (2004)

Laurienti, P. J., Burdette, J. H., Wallace, M. T., Yen, Y.-F., Field, A. S. & Stein, B. E. Deactivation of sensory-specific cortex by cross-modal stimuli. *Journal of cognitive neuroscience* 14, 420-429 (2002)

We have provided specific answers to each of the reviewers' comments below (Our responses are in blue). The new text in the manuscript and supplementary information file is shown in red.

REVIEWER COMMENTS

Reviewer #1 (Remarks to the Author):

Lohse and colleagues describe new subcortical pathways by which whisker stimulation suppresses tone-induced neuronal responses in auditory thalamus and primary auditory cortex. The authors show that whisker stimulation takes effect through primary somatosensory cortex via the auditory midbrain and auditory thalamus.

This is an interesting and impressive study with an incredible amount of data. I have only few comments, most of them minor.

Thank you.

1) My main criticism is that the authors do not show directly that the pathway through the IC shell is conveying whisker-induced suppression of tone responses in auditory thalamus and cortex. They instead show that stimulating IC cells receiving S1 input can suppress thalamic responses. It would strengthen the conclusions if the authors could silence the IC shell during tone and whisker stimulation and recordings in auditory thalamus (and cortex) similar to the experiments in Figure 5a-c for S1 silencing. However, these experiments are very challenging and if the authors are not able to perform them, in my opinion the study is strong enough without these data, but the authors should mention this pitfall in the discussion.

We agree that this is a worthwhile experiment but, as the reviewer acknowledges, it is also a very challenging one and despite our best efforts we have not been able to demonstrate effective silencing of the S1-recipient cells in the IC (i.e., we have been unable to electrophysiologically isolate cells in the LCIC that receive inputs from S1 in order to test whether these cells can be silenced optogenetically during whisker stimulation). This is likely due to a) the distributed nature of S1-recipient cells in IC, and b) insufficient labelling of these cells (due to limits in AAV1-hsyn-cre trans-synaptic transfection efficacy).

By demonstrating an essential role for S1 in whisker-induced suppression of auditory responses and, at the same time, ruling out both auditory cortex and TRN in mediating somatosensory suppression of the auditory thalamus, we believe we have excluded any other major potential source of inhibition except the IC. Nevertheless, we have changed the wording throughout the paper to highlight that we have shown that the S1-recipient IC neurons *can* account for the somatosensory suppression, and not that somatosensory suppression *necessarily depends* on these cells.

We have, however, performed an alternative set of experiments to address the reviewer's comment, the results of which provide additional evidence that S1 projects to inhibitory cells in the LCIC, which in turn project to the auditory thalamus. First, we carried out new electrophysiological recordings from the LCIC and show that a subset of cells found there are directly driven by whisker stimulation and produce enhanced auditory responses when whisker stimulation is delivered (new Supplementary figure 11). This is precisely the response profile expected from an excitatory projection from S1 to the LCIC. Second, we performed an anatomical experiment in which we used a transgenic VGAT-YFP-ChR2 mouse line where GABAergic cells, including those in the IC, are targeted with YFP. By viral trans-synaptic labelling of S1-recipient IC neurons in this mouse line, we were able to confirm the existence of GABAergic patches in the LCIC and found that S1-recipient neurons are predominantly located in these patches and that at least a substantial proportion of these neurons are GABAergic (VGAT+) (new Supplementary figure 13a,b). Finally, we have obtained additional anatomical evidence that S1-recipient IC cells project to the auditory thalamus (new Supplementary figure 13c).

Furthermore, the new experiments carried out to address reviewer 1 comment 2 (see below) constrain how S1 can causally (as has been shown by silencing of S1) contribute to somatosensory suppression of the auditory thalamocortical system.

2) Related to point 1, the authors do not show that direct connections from S1 to A1 do not convey at least part of the whisker-induced response suppression in A1. Again, experiments providing conclusive evidence for or against a contribution of direct cortical connections are very challenging and the study provides enough important new findings without these data, but the authors should discuss this more directly in the discussion section.

By silencing the auditory cortex, we demonstrated that the somatosensory suppression of MGBv neurons is not a result of circuitry running through A1, such as a direct S1 to A1 connection. However, extensive evidence exists for direct corticocortical connections and we appreciate the concern of the reviewer that A1 responses may nonetheless be modulated by circuitry involving direct inputs from S1. Based on previous studies, the most plausible corticocortical circuit capable of inducing suppression of sound-evoked activity would involve an excitatory projection from S1 to inhibitory interneurons in A1, akin to the projection from motor cortex that has been implicated in movement-induced suppression of sound-evoked cortical activity (Schneider et al., 2014, Nature 513, 189-194).

Our data show (as expected) that optogenetic activation of VGAT+ cells in A1 suppresses the spontaneous activity and auditory responses of most A1 neurons (Supplementary figure 7), so if the inhibitory interneurons are activated by whisker stimulation, this could provide the basis for the somatosensory suppressive effects. To address the reviewer's comment and to test whether S1 (or any other brain structure driven by somatosensory inputs) does indeed suppress auditory responses in A1 via activation of local inhibitory circuitry, we performed 2-photon calcium imaging of inhibitory interneurons in A1 while playing tones and stimulating the whiskers of awake mice (new Supplementary figure 1). We

found that inhibitory interneurons in A1 also display somatosensory suppression, rather than the facilitation expected for the typical cross-modal corticocortical projection. This therefore suggests that an equivalent S1-A1 projection to that described from motor cortex is unlikely to explain the somatosensory suppression of A1 auditory responses. The data from this experiment are described on lines 63-70 of the manuscript.

This, of course, does not rule out the possibility that direct cortico-cortical projections may have important roles under certain conditions, and we acknowledge this in the Discussion (lines 390-398, 399-400).

Minor:

3) It would be good to include some of the important numbers and stats in the manuscript text.

We have now moved all important statistics related to the supplementary figures into the main manuscript to aid the reader's appreciation of effects that are illustrated only in supplementary figures. We found that the readability of the results section suffers if statistical tests relating to the main figures are presented in the main text (instead of the figure legends). We would therefore prefer to keep the details of these tests in the main figure legends, as they are now and as is commonly the case in papers published by Nature Communications.

4) Line 152: the title "Somatosensory suppression of auditory cortex is inherited from the thalamus" is misleading. It is of course likely that this is the case since vMGB provides a major drive to A1, but the authors do not show this directly, only that thalamic neurons projecting to layer 1 of A1 show suppression of responses by whisker stimulation. The authors should change the section title accordingly.

We agree and have changed the title of this section to: *Auditory thalamocortical neurons are suppressed by whisker stimulation* (line 174).

5) Figure 3: is there a difference in the distribution of effects of whisker stimulation in the different medial subsections of the auditory thalamus? Also please indicate the location of the map in Figure 3j in the thalamic.

We originally investigated the SGN and MGBm/PIN separately. MGBm/PIN are difficult to separate anatomically, and these two nuclei have historically often been treated as one (Smith et al., 2006). The voronoi diagram in figure 3j further shows that SGN, MGBm, and PIN all display both facilitated and suppressed cells.

When analyzing SGN and MGBm/PIN separately, we found very similar results in these regions and therefore decided to analyze the medial sector of auditory thalamus as one when assessing somatosensory influences. Below we illustrate the data shown in figure 3 with

facilitated and suppressed units in SGN and MGBm/PIN analyzed separately (Rebuttal letter figure 1). In the interests of clarity, we would prefer to keep the existing Figure 3 as it is, i.e., with the data combined across the different medial subsections of the auditory thalamus. We have made clear in the text, however, that the results are very similar for the SGN and MGBm/PIN, justifying the analysis of data from these regions together (lines 134-135 and 168-172).

Rebuttal letter Figure 1. Units separately analyzed in SGN and MGBm/PIN showing somatosensory suppression or facilitation.

We have added the locations of the different auditory thalamic subdivisions to manuscript Figure 3j and have indicated with a dashed line the border between the lateral nuclei (MGBv and MGBd) and the medial auditory thalamic nuclei (MGBm/PIN/SGN). The voronoi diagram shows auditory thalamus collapsed in the rostro-caudal plane, making the positions of each subdivision approximate, so indicating where the exact borders lie is not possible (Fig 3j and lines 163-165).

6) Suppl. Figure 7: I do not see a decrease in the whisker-induced response suppression when silencing S1 in Suppl. Figure 7b.

The reviewer is correct. We made a mistake when reporting the statistics for this figure. The figure displays the correct values but the statistics reported in the figure legend were incorrect. The effect of S1 silencing on the medial sector of auditory thalamus only approaches statistical significance ($p = 0.07$). We apologize for this mistake and have corrected the manuscript accordingly (Lines 220-223 and Supplementary Fig. 8 legend). Importantly, this does not affect the message of the paper.

7) Do RBP4-positive neurons also project to IC, or a different population of layer 5 cells?

To address this question, we utilized the Allen Mouse Brain Atlas dataset, which contains injections in different transgenic mouse lines and in different areas of cortex (Rebuttal letter figure 2). In this dataset, we identified 3 mice with S1-bfd pan-neuronal labelling (Expt ids: 127866392, 112882565 and 100141473) and 3 mice with S1-bfd RBP4+ labelling (Expt ids: 272735030, 64825323 and 647806688) with reasonable injection sizes. We also included, for comparison, one mouse with RBP4+ labelling in the upper limb sector of S1 (S1-ul; Expt id: 26624948).

When comparing the projection patterns from S1-bfd RBP4+ cells vs pan-neuronal S1-bfd, we found that RBP4+ cells in S1-bfd have a very limited projection to the IC, mostly confined to its rostral sector (Rebuttal Figure 2b). This is in contrast to a denser projection along the rostrocaudal lateral shell of IC when labelling S1-bfd pan-neuronally (Rebuttal Figure 2a). Interestingly, the relative lack of RBP4+ cell projections to IC is not seen in all S1 regions. For example, RBP4+ cells in the upper limb sector of S1 have denser projections to IC than when the injection site is placed in the barrel field (Rebuttal letter figure 2c). This paucity of RBP4+ projection from S1 to IC may therefore be particularly pronounced in S1-bfd.

To illustrate and quantify the projection strength (Rebuttal Figure 2c) of S1 projections in each mouse, we used the “fluorescent projection volume” (i.e. area (mm^3) with fluorescent pixels (axons and terminal fields) in the area of interest, as estimated by the Allen Brain Institute) within each of the regions examined (central nucleus of IC (CNIC), dorsal cortex of IC (DCIC), lateral cortex of IC (LCIC), and thalamic posterior nucleus (PO)). To account for variation in injection sizes and visibility of axons, we normalized the projection strength in the IC subdivisions in each mouse to that of the higher-order somatosensory thalamic nucleus PO (Rebuttal letter figure 2d).

These analyses demonstrate that RBP4+ S1-bfd cells only account for a small fraction of the lateral IC shell projection from S1-bfd, and the example shown illustrates that the S1-bfd RBP4+ projection is confined to the rostral sector of IC. This very small projection to IC from RBP4+ cells in S1-bfd likely explains why optogenetic activation of S1-bfd RBP4+ cells does not induce suppression of activity in MGBv/d.

We have included this information in the legend to Supplementary figure 10.

a Pan-neuronal S1-bfd projection

b RBP4+ S1-bfd projection

d

Rebuttal letter Figure 2. a, Example of pan-neuronal S1-bfd projections to IC. b, Example of S1-bfd RBP4+ cell projections to IC. Images taken from the Allen Mouse Brain Atlas dataset. c, Normalized projection strength of pan-neuronal and RBP4+ cells from S1-bfd and S1-ul (upper limb) to CNIC, DCIC, and LCIC. d, Projection strength to PO (used for normalization of the injection size) for each mouse. Upper left values denote distance from lambda. Calibration bars: 560 μ m.

8) Figure 6c: please add information on the location of this image, e.g. by adding a square in Figure 6b.

As requested, we have added a square to panel b to show the location of the high magnification image in panel c.

9) Please add clearer images in Figure 7g to show cells in IC and location and density of axons in MGB.

The unclear labelling of cell bodies in Figure 7g is due to how the fused ChR2-YFP of AAV-ChR2-YFP actually labels cells, making it difficult to differentiate cell bodies from their axons and dendrites. We have kept the images in the main figure as they demonstrate the labelling with ChR2-YFP in a mouse in which we performed optogenetic activation experiments. But we also carried out additional experiments to produce images that allow us to address the reviewer's concern by clearly showing S1-recipient IC neurons (new Supplementary figure 13b), as well as the distribution of their axons in the MGB (new Supplementary figure 13c). See our response to the reviewer's point 1) above for details.

Reviewer #2 (Remarks to the Author):

In this manuscript, Lohse and colleagues report on the influence of the somatosensory system on auditory responses in midbrain, thalamus and cortex to study the details of integration of information across senses. This is an important addition to the literature as it is of paramount importance to shed light on our general understanding on perception.

Authors show that stimulation of the whiskers causes widespread suppression of sound-evoked activity in mouse primary auditory cortex. Authors further demonstrate that this suppression depends on the primary somatosensory cortex and is implemented through a descending circuit that links S1, via the auditory midbrain, with thalamic neurons that project to A1.

Overall I like the manuscript. The study is multidisciplinary and use several complementary techniques It is clearly written and easy to follow, so I have no major issues.

Perhaps the section on the discussion is a bit too long and authors could make an effort to trim down a bit so it is more focused and distinctive. I particularly like the subcortical emphasis that authors make as opposed to many other studies on cortex that tend to be corticocentric and to neglect the subcortical contribution to sensory processing. I also like the combination of awake and anaesthetized preparations. Again, I like the approach where authors give more importance to the anaesthetise, this is an appeal to the overall demand of awake preparations by many journals (including Nat Comm, as I have suffered of it myself). This is not to say I dislike awake, simply that it is not a mandatory experiment. Anyway.

So in general I am supportive for the study to be published in Nat Comm but I also have a few comments that authors may care to consider, or at least comment on it in the paper or rebuttal letter.

We are grateful for these positive comments on our study. Several of the responses required to the comments made by the reviewers required the addition of material to the Discussion. Nevertheless, we have gone through the Discussion carefully and reduced the existing text in this section by nearly 15% in length.

Specific comments

major

An important question, at least to me is that whilst authors have recorded from the CNIC and found no somatosensory-auditory interactions, this is not surprising at all. But I wonder what would happen if similar recordings are done in the non-lemniscal IC, i.e., in the lateral, dorsal and rostral regions surrounding the IC that are well known (as the authors themselves cite) to receive somatosensory inputs. This possibility needs to be further explored, ideally with additional experiments or at the very least, this limitation should be discussed.

We agree that this is an important point, which we have addressed by performing two new recording/imaging experiments that characterize the responses of neurons in the lateral and dorsal shell of the IC.

First, we performed electrophysiological recordings in the lateral shell of the IC while playing tones and stimulating the whiskers. Given the existence of clustered GABAergic neurons in the lateral cortex of the IC that receive inputs from the somatosensory cortex (Lesicko et al., 2016, J. Neurosci 36, 11037–11050) and project to the auditory thalamus (Supplementary figure 13), we hypothesized that whisker stimulation should modulate the responses of neurons in the LCIC. Our recordings confirmed this hypothesis. We found that subsets of frequency-tuned neurons ($n = 94$, 2 mice) in the lateral cortex of the IC were driven by whisker stimulation alone (17%, $P < 0.05$, t -test) and/or had their responses to sound facilitated by whisker stimulation (6.4%, $P < 0.05$, two-way t -test). Interestingly, the auditory responses of another subset of units in the lateral cortex of the IC were significantly suppressed

by somatosensory inputs (9.6%, $p < 0.05$, t -test). These new findings have now been added to the main text (lines 344-348) and are illustrated in the new Supplementary figure 11.

To assess the influence of somatosensory stimulation on sound-evoked activity in the dorsal cortex of IC (DCIC), we performed two-photon calcium imaging in awake mice. As the DCIC is very superficial and tends to be very thin, this approach is better suited than electrophysiology to record neural activity in this brain region. To further ensure that we avoided the central nucleus, we used a labelling approach that targets the calcium indicator predominantly to the shell of the IC, i.e. we injected an anterogradely transported AAV1-hSyn-cre virus into the auditory cortex of GCaMP6f reporter mice. We found that, overall, cells in the dorsal cortex were suppressed by whisker stimulation. These new findings from this 2-photon imaging experiment have now been added to the new Supplementary figure 12 and are described on lines 348-356.

Together, we hope these new experiments meet the reviewer's request to provide a more comprehensive characterization of the influences of whisker stimulation on the shell (lateral cortex and dorsal cortex) of the IC.

Another important issue is the data from MGB and particularly that, shown in supplementary fig 4. Authors state: 'Similar somatosensory influences on auditory responses were found in MGBv/d of awake, head-fixed mice (Supplementary Fig. 4).' However, I find that although statistically significant it is very marginally and wonder the functional significance of this strong statement.

The reason for stating that 'Similar somatosensory influences on auditory responses were found in MGBv/d of awake, head-fixed mice' is that in both awake and anesthetized preparations we observed significant somatosensory suppression in MGBv/d, with stronger suppression at the BF than at the tails of the frequency response profiles. However, we can understand why the reviewer questioned this.

When studying somatosensory-auditory interactions in awake mice, there is a possibility that the data might be affected by potential confounding influences from uncontrolled sources, such as movements initiated by the mouse when the whiskers are deflected or arousal changes (or other behavioural state changes) during auditory or somatosensory stimulation. Indeed, a recent study presented at the 2020 Neuromatch conference by Bimbard et al. (https://neuromatch.io/abstract/?submission_id=recFXjTDCKa4MgFDB) highlighted the issue of possible movement confounds in multisensory studies in awake animals, by showing that auditory influences on visual cortex arise to a large degree because of movements that are triggered by the auditory stimuli. Our anesthetized preparation avoids this issue, enabling us to have greater confidence in the basis for the comparable somatosensory influences on auditory thalamus and cortex observed in awake animals.

Furthermore, the 80 dB SPL sound stimulus is quite loud for an awake mouse (though standard and appropriate for anesthetized preparations), and is close to the thresholds for triggering the startle reflex (e.g., Parham and Willott (1988); Plappert et al. (2001); Chambers et al. 2016)). Therefore, it is possible that in awake mice the 80 dB SPL tones themselves trigger a

small movement of the whiskers, which might make the passive whisker stimulation we provide less salient. Based on the published thresholds for triggering the startle reflex, this should not be an issue for tones presented at 60 dB SPL. We included data obtained at both sound levels in order to show that the somatosensory effects were still present in awake mice when we used the same sound levels as in our anesthetized preparation.

Another factor is that neurons in the MGB and auditory cortex are likely to produce stronger responses to tones presented at the same sound level in awake than in anaesthetized animals. We mention this because the somatosensory suppression is substantially stronger in awake animals at 60 dB SPL compared to 80 dB SPL. This could be because the responses recorded at the higher sound level are closer to the ceiling on the neurons' input-output response curve, which would lead to suppressive effects appearing to be smaller at stronger auditory drive - as would be the case at 80 dB SPL compared to 60 dB SPL tone presentations.

We would maintain that we do see comparable effects of whisker stimulation on auditory responses in the MGB of awake and anesthetized mice, but there are several reasons why the magnitude of the effects may differ in the two preparations. Importantly, Supplementary Fig. 5 (previously Supplementary Fig. 4) shows the same sound-level dependence in the magnitude of somatosensory suppression of auditory responses in the MGB and A1, adding weight to our conclusion that multisensory interactions in the thalamus most likely provide the basis for the effects seen in the auditory cortex.

However, to address the reviewer's comment, we have dropped the word 'similar' when comparing the data from awake and anesthetized mice and rewritten this section of the results (lines 85-91). We have also added additional text about the sound-level dependence of the suppression to the legend to Supplementary Fig. 5.

In connection with this issue, authors later on study the medial part of the MGB. So my big question here is that, either I have misunderstood, or authors have mix up lemniscal and non lemniscal MGB. The MGBv/d is analysed as a pool but in my humble view this is inappropriate. It would-be more reasonable to pool MGBd and MGBm, PP and PIL.... As these are the genuine non lemniscal regions. Then, perhaps the statistics are different, more robust and clear.

We certainly agree that MGBd is a non-lemniscal region and that MGBv is the lemniscal auditory thalamic region. This is stated on lines 101-103 in the Results and on line 408 in the Discussion. We combined data from MGBv and MGBd in the analysis simply because of the similarity in the effects of somatosensory stimulation on neuronal responses in these two areas. This is visible both in Figure 2e-g, where MGBv and MGBd are analyzed separately and in Figure 3j, which shows the distribution of modulation in units recorded across all of auditory thalamus. We therefore consider MGBv and MGBd as one functional module in relation to somatosensory influences. The medial nuclei MGBm/PIN/SGN on the other hand show diverse responses and modulatory effects of somatosensory inputs and therefore seem to work in a functionally distinct way from the lateral nuclei MGBv and MGBd. In this regard, it is worth noting that Smith et al. (2006, J Comp Neurol 496, 314–334) highlighted the similarity in the

physiological properties of neurons in the MGBv and MGBd versus those of neurons in more medial regions of the MGB.

We have now analyzed the effects of whisker stimulation on MGBv and MGBd separately for our awake data (Rebuttal letter figure 3). As in our anesthetized experiments, neurons in MGBv and MGBd were modulated by somatosensory inputs in similar ways. In MGBv, we observed significant suppression of the responses at BF at both 60 dB SPL ($P < 0.05$, $n = 141$ neurons) and 80 dB SPL ($P < 0.05$, $n = 141$ neurons). We sampled MGBd sparsely in these animals ($n = 16$ neurons, 3 mice) and again observed numerically weaker BF responses in the somatosensory-auditory condition (particularly at 60 dB SPL); our small sample size for this region is likely to be the reason why this effect was not significant ($p > 0.05$, $n = 16$) in awake mice. As shown in Figure 2, we did find significant suppression of both BF and noise responses in separate analyses in MGBv and MGBd in our larger sample sizes from anesthetized mice. On the other hand, this was not the case in the MGBm/PIN/SGN, suggesting that MGBd is distinct from these medial regions in terms of its somatosensory inputs.

We have explained the rationale for combining the data from different regions of the auditory thalamus in the Results (lines 106-108, 134-135 and lines 168-172) and the legend to Figure 5 (lines 237-239). The study by Smith et al (2006) (reference 63) is also cited in the Discussion (lines 433-435).

Rebuttal letter figure 3. Somatosensory suppression, using 60dB SPL or 80 dB SPL tones in awake mice, with MGBv and MGBd analyzed separately.

The fact that responses in the medial MGB regions are more heterogeneous does not mean are less important or functionally significant, and this should be further explored and discussed.

We agree that the heterogenous somatosensory modulation of responses in the medial MGB does not mean it is less important. As requested, we added more information about the data

recorded here throughout the Results section (including the addition of new data from awake mice on lines 145-147), and slightly expanded the paragraph in the Discussion (beginning on line 420) that deals with the properties and targets of these neurons. It is important to stress, however, that the starting point for this study was the modulatory effect of whisker stimulation on the responses of neurons in A1 and our overall aim was to investigate the circuitry underpinning the suppressive cross-modal interactions observed there.

Minor

The first sentence of the results could go to the end of intro, or at least it would be very informative to state right at the end of Intro what are you going to do. I miss a specific aim. I think this would be very helpful for a general audience.

We now state in the introduction - immediately prior to the summary of results - that our aim was to investigate the role of subcortical circuits in shaping multisensory processing in the auditory cortex (lines 40-41).

Although authors refer to Figure 1 b and c. as tuning curves, strictly speaking they are not so, I would rather refer to them as isointensity functions that reflect the BF.

The reviewer is quite right and we have replaced the term 'tuning curves' by 'frequency response profiles' throughout the manuscript and supplementary material. We think this term is more intuitive than isointensity functions.

I wonder if the panel from fig 2a-c could be combined with wig 1, so the reader can directly compare the AC and MGB,

Figure 1 illustrates and summarizes all the data from A1 in awake mice, whereas Figure 2 provides a comparison of the responses in A1, MGBv and MGBd in anaesthetized mice to illustrate the very similar effects of somatosensory stimulation in each region (with corresponding data from the cortex and thalamus of awake mice shown in Supplementary figure 5). The organization of the figures therefore already allows a direct comparison of AC and MGB. We may have misunderstood what the reviewer was suggesting, but we think the present arrangement illustrates the points we wanted to make more effectively.

Also, I think it would be very valuable if authors include the MGB borders in the Voronoi diagram from fig 3j.

As requested, we have added the locations of the different auditory thalamic subdivisions to the voronoi diagram in Figure 3j, and have indicated with a dashed line the border between the

lateral nuclei (MGBv and MGBd) and medial auditory thalamic nuclei (MGBm, PIN and SGN). We also point out in the legend that the voronoi diagram shows the auditory thalamus collapsed in the rostro-caudal plane. Because of this, the positions of the subdivisions are approximate, so drawing in exact borders is not possible.

I encourage the authors to engage in scientific discussion with me and other reviewers, nothing of what I say here is a must, I just try to be critical but constructive take on this lovely work you have done

Thank you for this generous compliment. We have addressed every comment made by all three reviewers, in most cases by adding new data and/or analyses to comply with those suggestions.

Reviewer #3 (Remarks to the Author):

In this manuscript, Lohse et al. first describe a divisive normalization process in the auditory cortex (AC) when sound and whisker stimuli are presented simultaneously. They then show that this multisensory modulation occurs in the dorsal and ventral MGB (dMGB, vMGB) and not in the central nucleus of IC (CNIC). In contrast to the dMGB and vMGB, mMGB neurons show a mix of facilitation and inhibition by touch. The authors go on to show that the diminished responses in dMGB and mMGB are not driven by the auditory portion of the thalamic reticular nucleus (TRN). Finally, they use trans-synaptic activation of presumed inhibitory neurons from shell regions of IC and show that activation of these neurons mimics the same suppression seen by multimodal stimulation.

This work makes an important contribution to our understanding of multimodal integration. Indeed, it postulates a very novel hypothesis that long-range somatosensory projections to the shell IC are important for suppressive auditory-somatosensory interactions and provides evidence to support this hypothesis. The manuscript is well-written and illustrated.

Thank you.

There are some weaknesses that should be addressed:

1. Do the RBP4+ cells in layer 5 of somatosensory cortex not project to the IC? Given mounting data that suggest that layer 5 cortico-thalamic cells branch to the midbrain, one would assume that these neurons also project to IC. If so, one would expect that their activation would engage the inhibitory neurons in the shell IC, as postulated in Fig 8, and suppress multimodal responses in the MGB (and AC). Please comment on this.

This same point was raised by reviewer 1. Please refer to our response to her/his comment #7 for a full answer. We were also expecting that optogenetic activation of S1-bfd RBP4+ cells would induce suppression of activity in MGBv/d cells. The reason why this was not the case seems to be that the projection from RBP4+ neurons in S1-bfd to the IC is very small (unlike other neurons in S1-bfd and RBP4+ neurons in other parts of S1, which do project more profusely to the IC). We now refer to the relevant data in the Allen Mouse Brain Atlas that illustrate this in the legend to Supplementary figure 10.

2. The authors speculate that inhibitory neurons from the shell IC targeted by the somatosensory cortex provide ascending inhibition to the MGB. Another possibility is that shell IC neurons (excitatory or inhibitory) could engage local circuits at the level of the IC that then modulate the inhibitory ascending projections to the MGB. To that end, it would have been interesting to identify the target IC neurons as being excitatory or inhibitory and the degree to which they project locally. The authors should modify their model in Fig 8 to account for the potential that somatosensory cortex engages local circuits that then send a projection to the MGB. Also, why are no cell bodies seen in the image in 7G?

This is a very good point, which we have addressed by performing new anatomical experiments in which we achieved anterograde trans-synaptic labeling of S1-recipient IC cells in VGAT-ChR2-YFP mice. This revealed that neurons in the IC that receive input from S1 are GABAergic (VGAT+) and primarily distributed in the GABAergic patches in the lateral cortex of the IC (Supplementary figure 13a,b). Furthermore, we have provided additional anatomical evidence that S1-recipient IC neurons project to the auditory thalamus (Supplementary figure 13c). Finally, it is evident from the images produced by these experiments that S1-recipient IC neurons do not exclusively project to the thalamus but also project locally as well as more widely within the IC, and in particular within the shell of the IC. The new data are described on lines 339-366) and we have modified Figure 8 to highlight the presence of facilitatory and suppressive interactions in the IC shell.

The unclear labeling of cell bodies in Figure 7g is due to how the fused ChR2-YFP of AAV-ChR2-YFP actually labels cells, making it difficult to differentiate cell bodies from their axons and dendrites, but the cell bodies are also labeled (see Supplementary figure 13b). See for comparison an example in Supplementary Figure 1 of Shabel et al. (2012) Neuron 74, 475-481.

3. Were the TRN data obtained in awake animals? If not, the authors should comment that modulating the TRN in an anesthetized animal is unlikely to account for its role in multisensory processing given the strong modulation of this nucleus by arousal. In addition, the authors should specify that the two animals showing the mild effects of TRN stimulation (fig 7C) were the same animals in Figs 7e and f. Also, was there any relationship between the degree of modulation that any one cell had in terms of TRN general effects vs. TRN multimodal effects? If

there is a linear relationship, then the negative results may be explained by a large subset of cells whose TRN inputs were not adequately silenced.

The TRN optogenetic silencing experiments were carried out in anesthetized mice. Although we agree that TRN is known to be modulated by arousal state (including awake vs anesthetized), we believe it is valid to assess the role of TRN in establishing somatosensory suppression of MGB using optogenetic methods in anesthetized animals, as somatosensory suppression of MGN neurons is robustly present under anesthesia. Our results show that TRN does not appear to establish the somatosensory suppression observed in the MGB. However, it is, of course, possible that TRN may have an additional role in somatosensory modulation in awake – and particularly behaving – animals. We now acknowledge this on lines 311-315).

The data shown in Figure 7c-f come from the same neurons in the same two animals (this is now stated explicitly in the figure legend, line 322). In relation to the question of whether there exists a relationship between the TRN general effect vs TRN modulation of somatosensory suppression, we find no support for the hypothesis that silencing TRN affects somatosensory modulation more in cells that were strongly modulated by TRN in general. In the figure below (Rebuttal letter figure 4), we have plotted the relationship between the strength of general TRN modulation (auditory response with concurrent optogenetic silencing of TRN divided by auditory response alone) versus TRN-dependent modulation of somatosensory suppression. The correlation between TRN modulation of the tone response at BF vs TRN modulation of somatosensory suppression of the tone response was not significant (Pearson's $r = -0.055$, $p = 0.74$). Although we do not think that including this figure in the manuscript is necessary, we have added a statement summarizing this finding to the legend for Figure 7 (lines 325-327).

Rebuttal letter figure 4. (Lack of) correlation between strength of TRN silencing of BF tone responses and effect of TRN silencing on somatosensory suppression in individual units in MGBv/d.

4. Imaging thalamocortical terminals in layer 1 will miss the majority of thalamocortical terminals. Therefore, it is not appropriate to conclude that multimodal stimulatory signals are not relayed to the cortex. Therefore, Fig 4 adds very little to the manuscript. It is, however, appropriate to keep the thalamocortical terminal imaging in Fig 5.

We agree that it is possible that axons terminating in deeper layers of A1 may carry somatosensory facilitation of auditory responses or even standalone somatosensory drive from the medial auditory thalamus. However, we believe that there are good reasons for arguing on the basis of Figure 4 that these multisensory signals are not relayed from auditory thalamus to A1. Although layers 3b/ 4 are the main thalamic input layers from MGBv, both MGBv (showing somatosensory suppression) and MGBm (showing mainly somatosensory facilitation) project to layer 1, making layer 1 the most suitable layer to look for signals from these two regions of the auditory thalamus (Vasquez-Lopez et al., 2017, eLife 6, e25141). Furthermore, Vasquez-Lopez et al. (2017) study showed that much of the MGBv input to layer 1 is likely made up of branches of axons that also innervate the middle layers of the cortex where the majority of MGBv input to A1 terminates (Fig. 5d in Vasquez-Lopez et al., 2017). Measuring MGBv layer 1 thalamocortical input should therefore provide a good approximation of the input to the middle layers. While MGBm predominantly projects to cortical layer 1, it also projects to the infragranular layers.

There is therefore a possibility that thalamic cells facilitated by somatosensory stimulation project exclusively to the deep cortical layers, which would be missed by imaging thalamocortical axons in layer 1. If so, we would then expect our electrophysiological recordings across the different layers of A1 to show at least some deep layers neurons that were driven by whisker stimulation or that exhibited cross-modal facilitation, but this was not the case.

Together, we believe these findings indicate that A1 inherits auditory signals from the thalamus that are likely to be exclusively suppressed by whisker stimulation and that our imaging data from thalamocortical axon boutons in layer 1 is a key part of the evidence for this. We now explain the rationale for this (lines 181-194), whilst also acknowledging that we cannot rule out the possibility that somatosensory drive and facilitation from MGBm might be delivered exclusively to the deeper layers of A1 (lines 194-197). This point is also considered in the Discussion (lines 427-432).

Minor:

1. There are two references on auditory-somatosensory suppressive interactions that they authors may consider including:

Dehner, L. R., Keniston, L. P., Clemo, H. R. & Meredith, M. A. Cross-modal Circuitry Between Auditory and Somatosensory Areas of the Cat Anterior Ectosylvian Sulcal Cortex: A 'New' Inhibitory Form of Multisensory Convergence. *Cerebral Cortex* 14, 387-403, (2004)

Laurienti, P. J., Burdette, J. H., Wallace, M. T., Yen, Y.-F., Field, A. S. & Stein, B. E. Deactivation of sensory-specific cortex by cross-modal stimuli. *Journal of cognitive neuroscience* 14, 420-429 (2002)

Thank you for these suggestions. The study by Dehner et al. (2004) is certainly relevant and has been added to the Discussion (lines 391-394). We have not included the fMRI paper by Laurienti et al. (2002), which reports that visual stimulation reduces baseline activity but not auditory BOLD responses in auditory cortex (and vice versa), which is not straightforward to relate to the crossmodal changes in neuronal firing rate observed in our study and those carried out in other animal species.

Reviewer #1 (Remarks to the Author):

The authors addressed my concerns and I have no further comments

Reviewer #2 (Remarks to the Author):

Authors have done a nice revision and I have no further comments
good job

Reviewer #3 (Remarks to the Author):

The authors have done an excellent job at revising the manuscript and their main conclusions are now strengthened by the changes. I have two text revisions to suggest:

1. Lines 196-199: The last half of this sentence is too vague. Please consider changing "suggest that this is not the case" with "do not reveal evidence for multi-sensory facilitation in deep layers"
2. lines 316-320: It is fine to say that the TRN effects were observed in anesthetized animals. However, it is not appropriate to say that the effect occurred "independently of brain state," because only one brain state was used. Please modify the end of this sentence.

Nature Communications manuscript NCOMMS-20-34371A

Our responses to the reviewers' comments are indicated in blue with the new text in the manuscript in red.

REVIEWERS' COMMENTS

Reviewer #1 (Remarks to the Author):

The authors addressed my concerns and I have no further comments

Reviewer #2 (Remarks to the Author):

Authors have done a nice revision and I have no further comments
good job

Reviewer #3 (Remarks to the Author):

The authors have done an excellent job at revising the manuscript and their main conclusions are now strengthened by the changes. I have two text revisions to suggest:

1. Lines 196-199: The last half of this sentence is too vague. Please consider changing "suggest that this is not the case" with "do not reveal evidence for multi-sensory facilitation in deep layers"

We agree and have changed this sentence as follows (now lines 140-141):

Although we cannot rule out the possibility that MGBm axons carrying somatosensory drive and facilitation may terminate in the deep layers of A1, which were not imaged here, our electrophysiological data do not reveal evidence for multisensory facilitation in those layers (Figs. 1, 2; Supplementary Fig. 5).

2. lines 316-320: It is fine to say that the TRN effects were observed in anesthetized animals. However, it is not appropriate to say that the effect occurred "independently of brain state," because only one brain state was used. Please modify the end of this sentence.

This is correct, but we did not say that the TRN effects were observed independently of brain state. Instead, we stated that somatosensory modulation was observed independently of brain state, which is the case since recordings were made in both awake and anesthetized animals. Because the reviewer misread this sentence, we have clarified it as follows (now lines 202-204):

"Although we cannot rule out the possibility that TRN neurons may additionally contribute to crossmodal modulation in awake, behaving animals, our results suggest that they are not responsible for somatosensory suppression of neurons in MGBv/d, which occurs in both awake and anesthetized mice."